# Router-R1: Teaching LLMs Multi-Round Routing and Aggregation via Reinforcement Learning

**Haozhen Zhang**[1][*]    **Tao Feng**    **Jiaxuan You**
University of Illinois at Urbana-Champaign
{haozhenz,taofeng2,jiaxuan}@illinois.edu, [1]wazhz14@gmail.com

## Abstract

The rapid emergence of diverse large language models (LLMs) has spurred the development of LLM routers that assign user queries to the most suitable model. However, existing LLM routers typically perform a single-round, one-to-one mapping (*i.e.*, assigning each query to a single model in isolation), which limits their capability to tackle complex tasks that demand the complementary strengths of multiple LLMs. In this paper, we present **Router-R1**, a reinforcement learning (RL)-based framework that formulates multi-LLM routing and aggregation as a sequential decision process. Router-R1 instantiates the router itself as a capable LLM, leveraging its reasoning ability to interleave "think" actions (internal deliberation) with "route" actions (dynamic model invocation), and integrates each response into its evolving context. To facilitate learning, we employ a lightweight rule-based reward comprising format rewards, final outcome rewards, and a novel cost reward for optimizing the balance between performance and cost, opening a pathway toward enhancing performance-cost trade-offs via RL. Router-R1 also conditions only on simple model descriptors such as pricing, latency, and example performance, enabling strong generalization to unseen model selection. Experiments on seven general and multi-hop QA benchmarks show that Router-R1 outperforms several strong baselines, achieving superior performance while maintaining robust generalization and cost management.

 ulab-uiuc/Router-R1    Hugging Face Collection

## 1  Introduction

Large language models (LLMs) have proliferated at an unprecedented pace, with new architectures and fine-tuned variants released on a monthly or even weekly basis [21, 1]. To harness the complementary strengths of multiple LLMs (*e.g.*, one model's fluency versus another's factual accuracy), LLM routers have emerged as a critical infrastructure component, dynamically dispatching user queries to a single selected model to maximize answer quality and efficiency [3, 6, 2, 5, 26, 30, 4]. While this one-shot routing strategy can improve average performance, it overlooks the fact that truly complex reasoning tasks often require coordinated interactions among multiple models, orchestrating not just a single choice but a sequence of model calls to leverage their complementary strengths. This observation raises a key challenge: **how can we coordinate multiple LLMs in a multi-round routing and aggregation process to jointly solve complex tasks?**

Answering this challenge is non-trivial. **First**, the discrete decision process of selecting which LLM to call at each round is inherently non-differentiable, precluding straightforward end-to-end training via backpropagation. Although prior work has applied gradient-based methods to single-shot routing [3, 6, 2, 5], extending them to multi-round selection and aggregation rapidly becomes intractable.

---

[*]Work done as an intern at University of Illinois at Urbana-Champaign

39th Conference on Neural Information Processing Systems (NeurIPS 2025).

**Second**, existing routers operate in a single-step regime: given a query, pick one model, collect its output, and stop [3, 6]. However, complex tasks (*e.g.*, multi-hop question answering) demand a sequence of interleaved reasoning and model-selection decisions, and a one-off choice rarely suffices. Therefore, we must design an interplay mechanism that alternates between "long thought" reasoning (internal deliberation) and targeted LLM selection to refine an answer iteratively.

To address these challenges, we introduce **Router-R1**, a reinforcement learning–based framework for multi-round LLM routing and aggregation. Rather than making a single dispatch decision, we formulate LLM coordination as a sequential decision-making problem. At each step, Router-R1 chooses whether to perform internal reasoning ("think") or to invoke a specific model from a pool of available LLMs ("route"), gradually constructing an answer through iterative interaction. To support this interplay between reasoning and routing, we instantiate the router itself as a capable LLM, leveraging its inherent reasoning capability to perform long-form deliberation and targeted model selection. This flexible reasoning and model selection interleaving enables Router-R1 to adaptively compose the strengths of multiple LLMs in a task-aware manner. To optimize this decision policy, we adopt reinforcement learning and design a simple yet effective rule-based reward function composed of three parts: a format reward for producing well-structured outputs, a final outcome reward based on task correctness, and a cost reward that penalizes excessive use of expensive routed models, providing Router-R1 the capability to navigate performance–cost trade-offs during training. Additionally, Router-R1 exhibits strong generalization capability to newly added LLM candidates without the need for retraining by conditioning its routing decisions on simple descriptors such as pricing, latency, and example performance. Together, these components make Router-R1 a robust and flexible solution for coordinating multiple LLMs to solve complex reasoning tasks. Through comprehensive experiments on seven diverse QA benchmarks, covering both general and multi-hop question answering, we demonstrate that Router-R1 consistently outperforms several strong baselines and achieves state-of-the-art performance. Further analyses on cost-aware routing and generalization to unseen LLMs also highlight the flexibility and robustness of our approach.

Our contributions are summarized as follows:

- We propose **Router-R1**, a reinforcement learning–based framework for multi-round LLM routing and aggregation. By instantiating the router itself as a capable LLM, Router-R1 naturally interleaves internal reasoning and external model selection, enabling adaptive coordination across multiple LLMs for complex task solving.

- We design a simple and effective rule-based reward function consisting of format rewards, final outcome rewards, and cost rewards, enabling Router-R1 to navigate performance–cost trade-offs. Additionally, by conditioning on simple model descriptors, Router-R1 can generalize to unseen LLMs without retraining.

- Through extensive experiments on seven question-answering benchmarks, we show that Router-R1 outperforms several competitive baselines, achieving superior performance and robust generalization.

## 2   Related Work

### 2.1   Query-based Routers for LLM Selection

The rapid rise of various LLMs has spurred the development of query-based LLM routers, which aim to direct queries to the most appropriate model to improve response quality and efficiency. HybridLLM [5] proposes a dynamic router that selects between small and large LLMs based on predicted query difficulty and a user-defined quality budget. GraphRouter [6] frames LLM selection as inductive edge prediction over a task–query–model graph, enabling cost-performance estimation and effortless integration of new LLMs. To explicitly balance performance and cost, FrugalGPT [2] adopts an LLM cascade approach, while FORC [26] routes queries to appropriately sized models for cost-effective inference. TO-Router [30] unifies multiple domain-specific expert LLMs under one interface and dispatches queries based on task needs. C2MAB-V [4] leverages a cost-aware combinatorial multi-armed bandit to dynamically select optimal LLM subsets. RouterDC [3] enhances routing via dual contrastive learning between queries and LLM embeddings. Finally, RouteLLM [22] utilizes human preference data to dynamically choose between strong and weak LLMs, effectively reducing costs while maintaining quality.

In contrast to previous methods, Router-R1 treats routing as a sequential decision process, interleaving internal "think" steps with multi-round model routing to refine its answer by instantiating the router itself as a capable LLM. Moreover, its RL-based training harnesses a cost reward to balance performance and cost, enabling flexible performance–cost trade-offs and resource-aware routing.

## 2.2 Optimizing LLM Behaviors via Reinforcement Learning

In recent years, reinforcement learning (RL) has emerged as a powerful paradigm for fine-tuning large language models (LLMs) to better align with human preferences. Early work like RLHF (Reinforcement Learning from Human Feedback) [23], trains a reward model on human judgments and then applies policy-optimization algorithms such as PPO [27] to steer the base language model toward more desirable outputs. Building on this foundation, RLAIF [17] shows comparable or better performance on tasks like summarization and dialogue, and its direct variant (d-RLAIF) removes the need for an explicit reward model, improving efficiency. RRHF [37] ranks model-generated and external responses to train preference without a reward predictor, achieving results on par with RLHF. More recently, Direct Preference Optimization (DPO) series methods [25, 20] take a further step by training directly on human preference data, avoiding RL sampling and complex tuning while matching or outperforming RLHF. Alongside these advances in reward-driven fine-tuning, other RL-based techniques such as Search-R1 [13] have empowered LLMs to adaptively interact with external tools like search engines, allowing them to dynamically retrieve and incorporate external information during inference. These approaches highlight the potential of RL in optimizing LLM behaviors beyond static prompt-based generation, particularly in tasks requiring real-time information access and decision-making.

## 3 Router-R1

In this section, we detail our proposed Router-R1 into three parts. Section 3.1 introduces the reinforcement learning formulation with an LLM routing pool. Section 3.2 presents the reward curation strategy, which includes format rewards, final outcome rewards, and cost rewards. Section 3.3 describes the multi-round interaction training process, including training prompt template and multi-round interaction with an LLM routing pool. We show the architecture of Router-R1 in Figure 1.

### 3.1 Reinforcement Learning via Coordination with a LLM Routing Pool

In Router-R1, we adopt a general policy optimization objective with an LLM routing pool $\mathcal{P}$ formulated as:

$$\max_{\pi} \ \mathbb{E}_{x \sim D, \, y \sim \pi(\cdot | x; \mathcal{P})} \left[ r_\phi(x, y) - \beta \log \frac{\pi(y \mid x; \mathcal{P})}{\pi_{\text{ref}}(y \mid x; \mathcal{P})} \right], \tag{1}$$

where $\pi$ denotes the policy LLM to be optimized, and $\pi_{\text{ref}}$ is a reference LLM that may be fixed or updated iteratively for stable training. $x$ represents input samples from the dataset $D$, and $y$ denotes the generated outputs sampled from the policy $\pi_{\text{ref}}(y \mid x; \mathcal{P})$, which are interleaved with results obtained from accessing the LLM routing pool $\mathcal{P}$. $r_\phi(x, y)$ is the reward function, and $\mathcal{P}$ is the LLM routing pool, which provides a set of candidate LLMs available for selection. The KL regularization term ensures that the updated policy remains close to the reference, with the regularization coefficient $\beta$ controlling this trade-off. This formulation is general and encompasses various regularized reinforcement learning algorithms such as PPO [27], GRPO [28], and KL-constrained methods, allowing for flexible policy updates over a pool of LLM candidates.

Moreover, under this optimization objective, the policy LLM can dynamically select a candidate LLM from the LLM routing pool $\mathcal{P}$ multiple rounds, and obtain auxiliary information about the input sample by providing the candidate LLM with relevant context, thereby enhancing the reasoning process of the policy LLM. In this case, the policy LLM can be regarded as a coordinator, selecting and coordinating multiple candidate LLMs to jointly solve complex tasks.

### 3.2 Reward Curation

To provide Router-R1 with reasonable and effective supervision signals, we carefully design the reward function, including format rewards, final outcome rewards, and cost rewards, which we will describe in detail below.

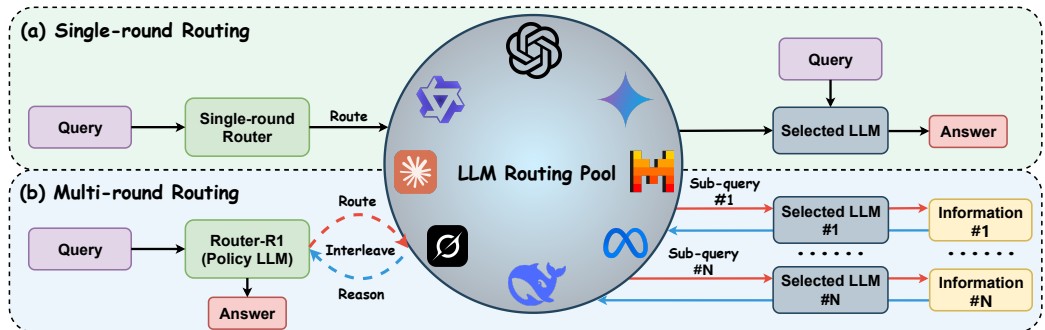

Figure 1: **Router-R1 architecture. (a) Single-round Routing**: A conventional router assigns each query to a single LLM in isolation via a one-shot decision, without internal reasoning or multi-model coordination. **(b) Multi-round Routing (ours)**: Router-R1 casts multi-LLM routing as a sequential decision process, which leverages an LLM-based router to interleave internal reasoning with external LLM routing and integrates retrieved information into its evolving context. This enables adaptive multi-model coordination for complex tasks, surpassing single-round routing with better performance.

### 3.2.1 Format Rewards

Inspired by [8], to stabilize the training and ensure that the LLM response rollouts conform to the predefined format (detailed below in Section 3.3), we impose strict format validation on Router-R1. Specifically, a format reward is assigned according to the following rule: if the response rollouts do not satisfy the required format, the format reward is set to -1; otherwise, it is set to 0:

$$\mathbf{R}_{\text{format}} = \begin{cases} -1, & \text{if the format is incorrect} \\ 0, & \text{if the format is correct} \end{cases} \tag{2}$$

### 3.2.2 Final Outcome Rewards

In Router-R1, we adopt Exact Match (EM) to measure the correctness between the answer predicted by LLM and the ground truth, and utilize it as the only final outcome reward to guide the optimization of Router-R1:

$$\mathbf{R}_{\text{outcome}} = \mathbf{EM}(y_a, g_t), \tag{3}$$

where $y_a$ is the predicted answer extracted from the generated output $y$ and $g_t$ denotes the ground truth. EM emphasizes the complete and strict matching of the predicted answer with the golden answer, which has been proven by many works to be a concise and effective rule-based reward [8, 13].

### 3.2.3 Cost Rewards

To balance the additional cost of invoking a candidate LLM from the routing pool, we introduce the computational cost incurred by querying a candidate LLM as a cost reward. This design enables Router-R1 to potentially optimize not only for model performance but also for the trade-off between performance and computational efficiency.

Formally, the cost reward is inversely proportional to both the number of output tokens produced by the candidate LLM and the model-dependent cost-per-token function, which reflects the computational price of using different models:

$$\mathbf{R}_{\text{cost}} \propto -m(P_{\text{LLM}}) \cdot T_{\text{out}}, \tag{4}$$

where $P_{\text{LLM}}$ denotes the number of parameters of the selected candidate LLM and $T_{\text{out}}$ is the number of output tokens generated by it. $m(\cdot)$ is a predefined cost function that maps model size to its per-token computational cost (*e.g.*, based on pricing tiers of the LLM API service). Note that the cost reward will be normalized to between 0 and 1 during training.

In this way, the larger the model size, the more output tokens, and the smaller the cost reward, which provides Router-R1 with the capability to achieve performance-cost balance.

### 3.2.4 Overall Rewards

To sum up, the overall reward function of Router-R1 can be defined as:

$$r_\phi(x, y) = \mathbf{R}_{\text{format}} + (1 - \alpha)\mathbf{R}_{\text{outcome}} + \alpha\mathbf{R}_{\text{cost}} \tag{5}$$

where $\alpha$ serves as a hyperparameter controlling the balance between model performance and cost.

In particular, to mitigate reward hacking [29] and improve optimization stability, we introduce a hierarchical reward as a refinement to the overall reward function (not shown in Formula 5 for brevity). In Router-R1, the three reward components are assigned different priorities, decreasing from left to right in Formula 5. Concretely, if the format reward is -1, the remaining two rewards are nullified (set to zero), regardless of their original values. This hierarchical design enforces critical constraints before optimizing for performance or computational efficiency, contributing to the stable and reliable training of Router-R1.

In our experiments, such a clear rule-based reward combination is sufficient to optimize Router-R1 well, which also demonstrates the rationality and effectiveness of our reward function design.

## 3.3 Multi-Round Interaction Training Paradigm

In this section, we describe the training prompt template of Router-R1 and the multi-round interaction with the LLM routing pool.

### 3.3.1 Training Prompt Template

Inspired by [8, 13], we construct the training prompt template for Router-R1 as shown in Figure 2. To ensure accurate and well-justified responses, we adopt a structured prompt combining internal reasoning and selective external querying. Upon receiving a question, Router-R1 first performs internal analysis within a `<think>` and `</think>` block to assess whether additional information is required. If so, it queries a suitable specialized LLM via `<search>` Candidate LLM: Query `</search>` based on predefined LLM routing pool and model descriptions that include details such as parameter size and task specialization (we detail it in Appendix C. ). Retrieved information is returned within `<info>` and `</info>` tags, and the process may iterate to gather complementary insights. The final answer is output within an `<answer>` and `</answer>` block.

It is worth noting that the model descriptions here serve only as an initial prior for each candidate LLM. During the policy optimization process, the policy LLM adaptively learns the strengths and weaknesses of each candidate LLM through interaction and feedback. To further enhance adaptability, the prompt design also supports the seamless integration of new candidate LLMs without requiring retraining. Specifically, Router-R1 can achieve this generalization capability by incorporating the descriptions of newly added LLMs directly into the prompt. This flexibility allows Router-R1 to dynamically expand its routing pool and effectively accommodate the rapid and ongoing emergence of new LLMs.

---

Answer the given question. Every time you receive new information, you must first conduct reasoning inside `<think>` and `</think>`.
After reasoning, if you find you lack some knowledge, you can call a specialized LLM by writing a query inside `<search>` Candidate LLM: Query `</search>`.
Before each LLM call, you must explicitly reason inside `<think>` and `</think>` about "why external information is needed" and "which LLM from the list is most suitable for answering your query," based on the brief model descriptions provided below.
When you call an LLM, the response will be returned between `<info>` and `</info>`. You are encouraged to explore and utilize different LLMs multiple times to better understand their respective strengths and weaknesses, as well as gather more comprehensive information.
**Description of LLM Candidates: {candidates_intro}**
If you find that no further external knowledge is needed, you can directly provide your final answer inside `<answer>` and `</answer>`, without additional explanation or illustration.
Question: {question}

---

Figure 2: Training prompt template for Router-R1 (some texts are omitted for page space).

### 3.3.2 Multi-Round Interaction with a LLM Routing Pool

According to the prompt in Figure 2, Router-R1 first analyzes the input question to identify the necessary information and selects the most appropriate candidate LLM from a routing pool to query via a sub-question. During training, it learns to decompose complex queries and route them adaptively based on the strengths of different LLMs.

The routing process is triggered whenever a special `<search>` tag appears in the generated sequence, specifying the target LLM and intended sub-query. Upon detection, Router-R1 queries the designated LLM and inserts its response back into the sequence to continue reasoning (to ensure stable training, external responses marked by `<info>` tags are excluded from the loss computation). For complex tasks, Router-R1 can perform multi-round routing, iteratively integrating information from multiple sources to arrive at a final answer. For simple questions, Router-R1 can rely solely on the policy LLM's internal knowledge to produce answers, demonstrating its ability to judge whether external information is needed.

## 4 Experimental Setup

### 4.1 Datasets and Metrics

We evaluate Router-R1 on seven question-answering (QA) datasets, *i.e.*, **(1) General QA**: Natural Question (NQ) [16], TriviaQA [14], PopQA [19]; **(2) Multi-Hop QA**: HotpotQA (HpQA) [36], 2WikiMultiHopQA (2wiki) [9], Musique [32], and Bamboogle (Bamb) [24]. These datasets cover both single-hop and multi-hop QA benchmarks, providing a comprehensive testbed for evaluating the performance of Router-R1. For metrics, the correctness of the predictions generated by Router-R1 is evaluated using the Exact Match (EM) and F1-Score (F1) against the ground truth.

### 4.2 Baselines

To have a comprehensive evaluation, we compare our proposed Router-R1 with dozens of baselines:

- **Basic Baselines**. **(1)** Direct Inference (**Direct**), **(2)** Chain-of-Thought (**CoT**) Prompting [34], **(3)** supervised fine-tuning (**SFT**), **(4)** Retrieval-Augmented Generation, which utilizes Wikipedia-18 [15] as the external knowledge and E5 [33] as the retriever (**RAG**), **(5)** **Search-R1** [13].

- **Query-based LLM Routers**. These baselines follow a similar setup: one or more candidate LLMs are selected from the routing pool to answer the input question (or its sub-questions, unless specified), and their responses are integrated into the base model's input to generate the final answer, simulating Router-R1's response generation process for fair comparison. **(6)** Prompt the base LLM model to select a candidate LLM (**Prompt LLM**), **(7)** always select the largest LLM (**Largest LLM**), **(8)** KNN Router [10], **(9)** MLP Router [10], **(10)** BERT Router [22], **(11)** RouterDC [3], **(12)** GraphRouter [6], **(13)** prompt the base LLM to decompose the original question into sub-queries and assign each to a candidate LLM (**Prompt LLM***), **(14)** prompt the base LLM to decompose the original question into sub-queries and utilize KNN Router to assign each to a candidate LLM (**KNN Router***).

To train query-based LLM routers, we additionally construct a dedicated router training dataset. Specifically, each training question is independently fed to every model in the LLM routing pool multiple times with temperature sampling. The responses are then evaluated using the EM metric to assess answer quality. This process yielded, for each question, a set of EM scores corresponding to all candidate LLMs in the routing pool, effectively labeling each LLM with its performance on that question. These question–LLM score pairs form the supervision signal for training the router.

### 4.3 Implementation Details

We conduct our experiments using **Qwen2.5-3B-Instruct** [35] and **LLaMA-3.2-3B-Instruct** [7] as base models for training. To empower LLMs with multi-round routing capabilities, we design Router-R1 with a maximum of 4 routing steps per input query. The model is trained using veRL [2] for

---

[2] https://github.com/volcengine/verl

reinforcement learning in LLMs, employing the Proximal Policy Optimization (PPO) as the default algorithm. The batch size is set to 64, with a maximum of 225 training steps. The cost coefficient $\alpha$ is set to 0.0 in our main experiment unless otherwise specified.

To incentivize both single-round and multi-round routing capabilities during training, we construct a joint dataset consisting of 7K samples each from the NQ and HotpotQA datasets, respectively. This results in a 14K sample training set, which we find sufficient to induce effective routing strategies without requiring extensive data filtering or complex sampling procedures. As demonstrated in our experimental analysis in Section 5, this modestly sized dataset enables robust routing and aggregation behavior learning.

After training, we evaluate in-domain performance on NQ and HotpotQA datasets, where Router-R1 has seen similar data during training, and assess out-of-domain generalization performance across five other QA datasets mentioned above. For each evaluation dataset, we randomly sample 500 test instances (except for Bamboogle, which contains only around 120 test examples in total). All baseline models are trained (if needed) and evaluated under the consistent dataset and settings to ensure fair comparison. To support routing strategies, we adopt a diverse LLM routing pool comprising six representative models of varying sizes and families: **Qwen2.5-7B-Instruct** [35], **LLaMA-3.1-8B-Instruct** [7], **LLaMA-3.1-70B-Instruct** [7], **Mistral-7B-Instruct** [12], **Mixtral-8x22B-Instruct** [11], and **Gemma-2-27B-Instruct** [31]. The base model training is conducted on NVIDIA A6000 GPUs, while routing LLMs are accessed via NVIDIA NIM APIs[3]. More details about experimental details and prompts are described in Appendix A and C. We also provide a case study of Router-R1 in Appendix B.

## 5 Experimental Analysis

In this section, we present comprehensive empirical studies to evaluate the effectiveness of Router-R1. We analyze its performance across diverse QA benchmarks (Section 5.1), explore its adaptability to cost constraints (Section 5.2), test its generalization capability when encountering unseen LLMs (Section 5.3), and provide insights into its routing behavior and training dynamics (Section 5.4).

### 5.1 Main Results

We conduct comprehensive experiments across seven QA benchmarks, spanning both general QA and multi-hop QA settings, to evaluate the effectiveness of Router-R1 against a wide range of baselines. The results are presented in Table 1, from which we can derive some key observations.

**Router-R1 consistently outperforms all basic baselines across all seven datasets, achieving SOTA performance**. Compared to Direct, CoT, and SFT, which rely solely on the base LLM's internal knowledge and reasoning capabilities, Router-R1 delivers substantially better results, particularly on these knowledge-intensive tasks where those baselines often struggle. It also outperforms RAG by a significant margin, as Router-R1 can dynamically query specialized LLMs during multi-round reasoning, offering more flexibility and relevance than methods relying on static external retrieval. Among these baselines, Search-R1 stands out as a stronger method that supports multi-turn search engine calling, resembling Router-R1's multi-round routing capabilities. However, Router-R1 still shows clear advantages, achieving better results across both base LLM models and most QA datasets. Notably, Router-R1-Llama reaches the highest average exact match score of 0.409, while Router-R1-Qwen further improves this to 0.416. These results demonstrate the superior effectiveness and adaptability of Router-R1 in routing decisions for both general and multi-hop QA.

**Router-R1 significantly surpasses all LLM router baselines in overall performance, benefiting from its multi-round routing and interleaved reasoning**. Compared to single-round routers like Prompt LLM and KNN Router, Router-R1 shows clear and consistent improvements. Even when enhanced baselines like Prompt LLM* and KNN Router* are introduced, where inputs are decomposed into sub-queries before routing, Router-R1 maintains a significant advantage. This superiority stems from the core design of Router-R1: using an LLM as the router itself, enabling flexible interleaving of reasoning and routing steps. Such a design allows Router-R1 to adaptively coordinate across models and aggregate their strengths more effectively. Furthermore, Router-R1 consistently outperforms advanced LLM router baselines such as GraphRouter and RouterDC across

---

[3] https://build.nvidia.com/

Table 1: **Experimental results on seven QA datasets w.r.t. Exact Match**.

| Methods | General QA | | | Multi-Hop QA | | | | |
|---|---|---|---|---|---|---|---|---|
| | NQ[†] | TriviaQA | PopQA | HpQA[†] | 2wiki | Musique | Bamb | Avg. |
| **Qwen2.5-3B-Instruct** | | | | | | | | |
| Direct | 0.092 | 0.260 | 0.122 | 0.140 | 0.266 | 0.026 | 0.040 | 0.135 |
| CoT | 0.126 | 0.358 | 0.160 | 0.168 | 0.208 | 0.046 | 0.224 | 0.184 |
| SFT | 0.212 | 0.400 | 0.160 | 0.198 | 0.256 | 0.052 | 0.112 | 0.199 |
| RAG | 0.298 | 0.540 | 0.366 | 0.216 | 0.146 | 0.078 | 0.224 | 0.267 |
| Search-R1 | 0.328 | 0.510 | 0.324 | 0.236 | 0.278 | 0.090 | 0.272 | 0.291 |
| Prompt LLM | 0.300 | 0.580 | 0.340 | 0.268 | 0.262 | 0.108 | 0.448 | 0.329 |
| Largest LLM | 0.296 | 0.578 | 0.354 | 0.278 | 0.274 | 0.104 | 0.480 | 0.338 |
| KNN Router | 0.262 | 0.528 | 0.222 | 0.224 | 0.196 | 0.066 | 0.360 | 0.265 |
| MLP Router | 0.252 | 0.460 | 0.222 | 0.198 | 0.210 | 0.072 | 0.360 | 0.253 |
| BERT Router | 0.230 | 0.516 | 0.192 | 0.216 | 0.206 | 0.058 | 0.312 | 0.247 |
| RouterDC | 0.278 | 0.592 | 0.282 | 0.244 | 0.218 | 0.080 | 0.504 | 0.314 |
| GraphRouter | 0.276 | 0.586 | 0.280 | 0.234 | 0.180 | 0.076 | 0.448 | 0.297 |
| Prompt LLM* | 0.258 | 0.500 | 0.256 | 0.206 | 0.248 | 0.078 | 0.472 | 0.288 |
| KNN Router* | 0.236 | 0.478 | 0.232 | 0.154 | 0.234 | 0.072 | 0.384 | 0.256 |
| **Router-R1-Qwen** | **0.388** | **0.706** | **0.384** | **0.352** | **0.434** | **0.138** | **0.512** | **0.416** |
| **Llama-3.2-3B-Instruct** | | | | | | | | |
| Direct | 0.202 | 0.328 | 0.176 | 0.144 | 0.134 | 0.018 | 0.048 | 0.150 |
| CoT | 0.256 | 0.468 | 0.182 | 0.172 | 0.168 | 0.040 | 0.272 | 0.223 |
| SFT | 0.076 | 0.098 | 0.084 | 0.100 | 0.224 | 0.026 | 0.016 | 0.089 |
| RAG | 0.308 | 0.478 | 0.356 | 0.162 | 0.084 | 0.038 | 0.176 | 0.229 |
| Search-R1 | 0.372 | 0.578 | 0.360 | 0.282 | 0.226 | 0.084 | 0.272 | 0.311 |
| Prompt LLM | 0.304 | 0.638 | 0.374 | 0.248 | 0.198 | **0.132** | **0.528** | 0.346 |
| Largest LLM | 0.344 | 0.616 | 0.394 | 0.258 | 0.242 | 0.122 | 0.472 | 0.350 |
| KNN Router | 0.292 | 0.572 | 0.254 | 0.210 | 0.182 | 0.078 | 0.376 | 0.281 |
| MLP Router | 0.282 | 0.506 | 0.248 | 0.178 | 0.188 | 0.064 | 0.360 | 0.261 |
| BERT Router | 0.256 | 0.560 | 0.222 | 0.210 | 0.188 | 0.066 | 0.296 | 0.257 |
| RouterDC | 0.310 | 0.614 | 0.298 | 0.250 | 0.204 | 0.088 | 0.504 | 0.324 |
| GraphRouter | 0.316 | 0.602 | 0.290 | 0.222 | 0.170 | 0.084 | 0.416 | 0.300 |
| Prompt LLM* | 0.236 | 0.446 | 0.164 | 0.118 | 0.080 | 0.036 | 0.208 | 0.184 |
| KNN Router* | 0.202 | 0.398 | 0.166 | 0.096 | 0.060 | 0.030 | 0.176 | 0.161 |
| **Router-R1-Llama** | **0.416** | **0.680** | **0.432** | **0.322** | **0.368** | 0.128 | 0.520 | **0.409** |

**Bold** indicates the highest score in each column for each base model.

[†] indicates in-domain evaluation; all others are out-of-domain.

both base models and evaluation benchmarks. These improvements highlight the flexibility of Router-R1's routing mechanism, which interleaves internal reasoning with multi-round routing to progressively refine its answers. By adapting routing decisions to each query through interaction and feedback, Router-R1 demonstrates strong performance in general and multi-hop QA tasks.

**Router-R1 demonstrates strong generalization to unseen data**. Despite being trained with in-domain samples only on NQ and HotpotQA, it achieves robust performance on the remaining five out-of-domain datasets. This indicates that Router-R1 learns transferable routing and reasoning strategies from limited training data, underscoring its generalization ability across diverse QA tasks. More experimental results can be found in Appendix A.2.

## 5.2 Analysis of Cost Rewards

In this section, we investigate how varying the cost coefficient $\alpha$ influences Router-R1's learned performance–cost balance. We initiate the mapping function $m(\cdot)$ according to the pricing tiers of Together API[4] and evaluate four cost coefficients (*i.e.*, 0.6, 0.7, 0.8, and 0.9) on Router-R1-Qwen. The results, shown in Figure 3, reveal two clear trends. First, as the cost coefficient increases, overall performance gradually declines. Second, the cost reward rises simultaneously, indicating that the

---

[4] https://www.together.ai/pricing

policy places greater emphasis on minimizing expensive LLM calls. Notably, we observe that the integration of cost rewards encourages an emergent routing strategy where Router-R1 prefers to query smaller models first and only escalates to larger models when necessary, thereby forming an adaptive routing strategy that dynamically balances efficiency and accuracy based on the cost constraints.

These findings confirm that, by optimizing under a composite reward, Router-R1 dynamically adjusts its routing behavior to adhere to resource constraints and achieves a controllable trade-off between accuracy and computational expense. An extensive analysis of cost rewards and a case study are provided in Appendix A.2.2 and B, respectively.

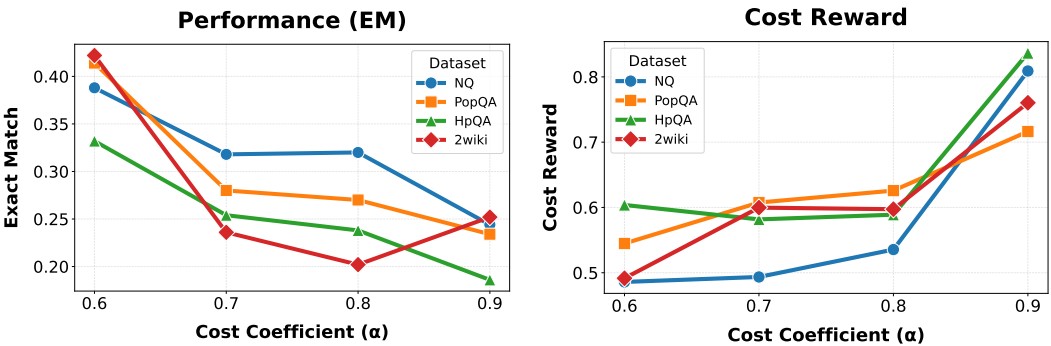

Figure 3: **Analysis of cost rewards on the NQ, PopQA, HotpotQA (HpQA), and 2WikiMulti-HopQA (2wiki) datasets.**

## 5.3 Generalization Capability to Unseen Candidate LLMs

To evaluate Router-R1's generalization ability, we extend the routing pool of Router-R1-Qwen with two previously unseen models, *i.e.*, Palmyra-Creative-122B[5] and LLaMA3-ChatQA-1.5-8B [18], and include their corresponding model descriptors in the inference prompt. Inference is conducted directly using the original Router-R1-Qwen checkpoint without any additional fine-tuning.

The results, presented in Table 2, show that Router-R1-Qwen maintains and even slightly improves performance across all four benchmarks when augmented with the new candidate LLMs. In particular, it achieves new best scores on several datasets, including TriviaQA, PopQA, and overall averages. This improvement demonstrates Router-R1's capacity to generalize to unseen candidate LLMs by interpreting their textual descriptors at inference time. Rather than relying on prior exposure or retraining, Router-R1 leverages descriptor-based reasoning to infer the strengths of new models and dynamically assign queries accordingly. Notably, this is achieved without sacrificing in-domain performance on tasks like NQ and HotpotQA, further validating the robustness of Router-R1's routing strategy. Compared to baselines, Router-R1 demonstrates stronger generalization, achieving consistent improvements with newly added LLMs, while baselines like GraphRouter and Prompt LLM* show limited or inconsistent gains, highlighting their weaker adaptability to unseen models.

Overall, the results highlight Router-R1's ability to flexibly adapt to unseen candidates and selectively utilize newly introduced LLMs, indicating promising potential for real-world deployment in evolving model ecosystems.

## 5.4 Discussion

**LLM API Call Count Analysis**. To assess the adaptability of Router-R1 to tasks of varying difficulty, we analyze the average number of LLM API calls (*i.e.*, the number of times candidate LLMs within the routing pool are invoked) by Router-R1-Qwen across seven QA benchmarks. As shown in Figure 4a, Router-R1-Qwen makes significantly more average LLM API calls on multi-hop QA benchmarks (*i.e.*, HotpotQA, 2WikiMultiHopQA, Musique, and Bamboogle) compared to general QA benchmarks (*i.e.*, NQ, TriviaQA, and PopQA). This indicates that Router-R1 can adaptively

---

[5] https://build.nvidia.com/writer/palmyra-creative-122b

Table 2: **Experimental results of generalization capability of Router-R1 on NQ, TriviaQA, PopQA, and HotpotQA (HpQA) datasets w.r.t. Exact Match and F1-Score** ([‡] indicates a routing pool extension).

| Methods | NQ[†] | | TriviaQA | | PopQA | | HpQA[†] | | Avg. | |
|---|---|---|---|---|---|---|---|---|---|---|
| | EM | F1 | EM | F1 | EM | F1 | EM | F1 | EM | F1 |
| Prompt LLM | 0.300 | 0.437 | 0.580 | 0.694 | 0.340 | 0.409 | 0.268 | 0.407 | 0.372 | 0.487 |
| Prompt LLM[‡] | 0.296 | 0.433 | 0.584 | 0.697 | 0.342 | 0.412 | 0.266 | 0.409 | 0.372 | 0.488 |
| BERT Router | 0.230 | 0.362 | 0.516 | 0.635 | 0.192 | 0.251 | 0.216 | 0.335 | 0.289 | 0.396 |
| BERT Router[‡] | 0.238 | 0.370 | 0.550 | 0.662 | 0.236 | 0.323 | 0.198 | 0.322 | 0.306 | 0.419 |
| GraphRouter | 0.276 | 0.412 | 0.586 | 0.690 | 0.280 | 0.324 | 0.234 | 0.366 | 0.344 | 0.448 |
| GraphRouter[‡] | 0.282 | 0.420 | 0.594 | 0.696 | 0.276 | 0.322 | 0.228 | 0.360 | 0.345 | 0.450 |
| Prompt LLM* | 0.258 | 0.373 | 0.500 | 0.600 | 0.256 | 0.315 | 0.206 | 0.313 | 0.305 | 0.400 |
| Prompt LLM*[‡] | 0.212 | 0.311 | 0.440 | 0.541 | 0.196 | 0.247 | 0.166 | 0.252 | 0.254 | 0.338 |
| Router-R1-Qwen | **0.388** | 0.484 | 0.706 | 0.772 | 0.384 | 0.447 | **0.352** | 0.449 | 0.458 | 0.538 |
| **Router-R1-Qwen**[‡] | 0.382 | **0.493** | **0.722** | **0.778** | **0.402** | **0.464** | 0.346 | **0.459** | **0.463** | **0.549** |

**Bold** indicates the highest score in each column for each base model.

[†] indicates in-domain evaluation; all others are out-of-domain.

assess task difficulty and decide whether external LLM routing is necessary, demonstrating its ability to selectively utilize external resources when tasks are more complex.

**Convergence Analysis of Router-R1 Training**. To evaluate the convergence behavior of Router-R1, we show two crucial curves during its policy training: the reward and the policy LLM's action entropy. Figure 4b and 4c illustrate that Router-R1 converges within 100 training steps, as evidenced by rising rewards and decreasing policy entropy, indicating rapid and robust convergence. It's worth noting that occasional formatting errors may cause brief drops in reward, but our hierarchical reward design swiftly corrects them, ensuring stable and accelerated learning. In particular, we observe that without format rewards, Router-R1 exhibits greater training instability, frequently generating meaningless or nonsensical text that leads to severe formatting breakdowns in the output.

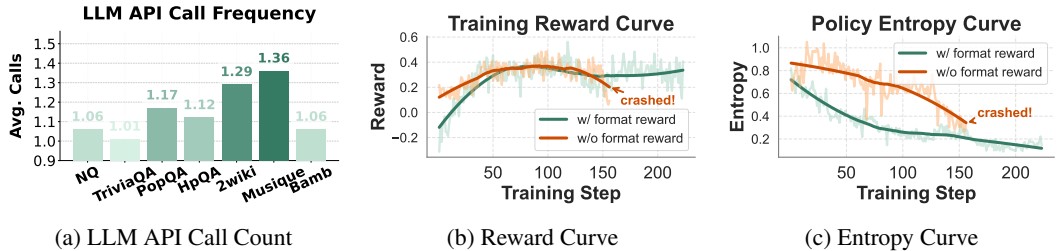

(a) LLM API Call Count     (b) Reward Curve     (c) Entropy Curve

Figure 4: **Analysis of LLM API call count and Router-R1 training convergence.**

## 6 Conclusion

In this paper, we introduce **Router-R1**, a reinforcement learning–based framework that formulates multi-LLM routing and aggregation as a sequential decision process. By instantiating the router as a capable LLM, Router-R1 interleaves internal reasoning with targeted model selection and incrementally builds responses through multi-round interaction. Our lightweight rule-based rewards, which combine format, outcome, and cost rewards, enable Router-R1 to achieve superior performance while learning flexible performance–cost trade-offs. On seven diverse QA benchmarks, it outperforms more than ten strong baselines and maintains robustness and generalization in the presence of distractor models. These results demonstrate the promise of RL-driven routing for orchestrating multiple LLMs.

## Acknowledgments

We sincerely appreciate the support from Amazon grant funding project #120359, "GRAG: Enhance RAG Applications with Graph-structured Knowledge", and research gifts from Meta and Lenovo.

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

# A   Experimental Details

## A.1   Implementation Details

### A.1.1   Model Descriptors

To enrich model descriptors and better inform routing decisions, we draw on the publicly available model cards[6]. Specifically, we prompt GPT-4o to generate brief, standardized summaries of each candidate LLM based on the information in these model cards. These concise descriptions are then injected into the training prompt of Router-R1. This enables the model to develop an initial understanding of each LLM's capabilities.

It is worth noting that these model descriptors serve solely as cold-start priors to guide early-stage decisions. During policy optimization, Router-R1 gradually refines its understanding of each LLM's strengths and weaknesses through repeated interaction and reward feedback. Furthermore, this design allows Router-R1 to seamlessly integrate new LLMs into the routing pool without retraining by simply appending their descriptions to the prompt. This flexibility supports fast adaptation to newly released models and enhances Router-R1's generalization in dynamic multi-LLM environments.

### A.1.2   Format Rewards

To ensure the structural quality and consistency of generated completions, we design a rule-based format reward that penalizes malformed outputs. The function inspects whether the completion correctly follows the expected structure using specific action tags. The format reward function imposes the following requirements:

- All tags must be properly opened and closed, with no nesting allowed.
- The response must begin with a `<think>` ... `</think>` block and end with a single `<answer>` ... `</answer>` block.
- At least one `<think>` ... `</think>` block must be present, and exactly one `<answer>` ... `</answer>` block is required.
- Each `<search>` ... `</search>` must be paired with a corresponding `<info>` ... `</info>` block.
- Routed queries must follow the format "llm_name: query", where the LLM name is valid and both parts are non-empty.

This format reward is lightweight but effective in enforcing disciplined structure in the model's output, guiding the policy toward well-formed and interpretable reasoning trajectories.

### A.1.3   Cost Rewards

To encourage cost-efficient routing decisions, we design a cost reward that penalizes the use of high-cost LLMs. This reward is computed as the inverse of a dynamically normalized cost value using a sliding window approach.

**Sliding Window Normalization.**   We maintain a fixed-size buffer of recent cost values (window size = 1000), and apply a smoothing transformation to each cost before normalization. Specifically, we adopt a square root transformation:

$$r' = \sqrt{r}$$

where $r$ is the raw cost associated with the selected LLM (*i.e.*, $r = m(P_{\text{LLM}}) \cdot T_{\text{out}}$).

To ensure robustness against outliers, we normalize $r'$ using the 5th and 95th percentiles of the buffer. Let $r_{\text{min}}$ and $r_{\text{max}}$ be the 5th and 95th percentile values respectively. The normalized cost is then computed as:

$$r_{\text{norm}} = \frac{r' - r_{\text{min}}}{r_{\text{max}} - r_{\text{min}}}$$

If the range $r_{\text{max}} - r_{\text{min}}$ is too small (less than a small threshold $\varepsilon$), we return a neutral reward value of $0.5$ to maintain stability.

---

[6] https://build.nvidia.com/models

**Reward Inversion.** To encourage the selection of low-cost LLMs, we invert the normalized value:

$$\mathbf{R}_{\text{cost}} = 1.0 - \text{clip}(r_{\text{norm}}, 0.0, 1.0)$$

This ensures that higher-cost models receive lower rewards, guiding the policy to prefer efficient candidates without relying on hard constraints. This cost reward design allows Router-R1 to learn cost-sensitive routing policies that adapt dynamically to the resource profiles of available LLMs.

**Discussion.** While the cost reward normalization stabilizes training, our experiments mainly focus on QA tasks. If Router-R1 were trained across heterogeneous domains (*e.g.*, math, code, QA), the large variation in response lengths could cause inconsistent cost scales even after normalization, complicating optimization. A promising direction for future work is to introduce task-level normalization to better align reward magnitudes across domains and improve multi-task training stability.

## A.2 More Experimental Results

### A.2.1 Main Results w.r.t. F1-Score

We present extensive experimental results on seven QA datasets with respect to F1-Score in Table 3. Router-R1 consistently outperforms all baselines across both general and multi-hop QA tasks, achieving the highest average F1-scores for both Qwen and LLaMA backbones.

Table 3: **Experimental results on seven QA datasets w.r.t. F1-Score.**

| Methods | General QA | | | Multi-Hop QA | | | | |
|---|---|---|---|---|---|---|---|---|
| | NQ$^{\dagger}$ | TriviaQA | PopQA | HpQA$^{\dagger}$ | 2wiki | Musique | Bamb | Avg. |
| **Qwen2.5-3B-Instruct** | | | | | | | | |
| Direct | 0.162 | 0.341 | 0.154 | 0.215 | 0.304 | 0.081 | 0.112 | 0.196 |
| CoT | 0.218 | 0.431 | 0.185 | 0.260 | 0.251 | 0.106 | 0.332 | 0.255 |
| SFT | 0.289 | 0.460 | 0.207 | 0.281 | 0.291 | 0.121 | 0.173 | 0.260 |
| RAG | 0.414 | 0.622 | 0.452 | 0.307 | 0.187 | 0.134 | 0.303 | 0.346 |
| Search-R1 | 0.407 | 0.575 | 0.383 | 0.328 | 0.317 | 0.145 | 0.387 | 0.363 |
| Prompt LLM | 0.437 | 0.694 | 0.409 | 0.407 | 0.393 | 0.205 | 0.579 | 0.446 |
| Largest LLM | 0.431 | 0.695 | 0.423 | 0.416 | 0.397 | 0.199 | 0.608 | 0.453 |
| KNN Router | 0.388 | 0.627 | 0.281 | 0.341 | 0.289 | 0.141 | 0.496 | 0.366 |
| MLP Router | 0.368 | 0.557 | 0.272 | 0.295 | 0.277 | 0.142 | 0.460 | 0.339 |
| BERT Router | 0.362 | 0.635 | 0.251 | 0.335 | 0.284 | 0.135 | 0.423 | 0.346 |
| RouterDC | 0.410 | 0.694 | 0.328 | 0.381 | 0.293 | 0.165 | 0.623 | 0.413 |
| GraphRouter | 0.412 | 0.690 | 0.324 | 0.366 | 0.258 | 0.154 | 0.546 | 0.393 |
| Prompt LLM* | 0.373 | 0.600 | 0.315 | 0.313 | 0.355 | 0.165 | 0.580 | 0.386 |
| KNN Router* | 0.360 | 0.572 | 0.272 | 0.248 | 0.299 | 0.138 | 0.493 | 0.340 |
| **Router-R1-Qwen** | **0.484** | **0.772** | **0.447** | **0.449** | **0.487** | **0.212** | **0.635** | **0.498** |
| **Llama-3.2-3B-Instruct** | | | | | | | | |
| Direct | 0.281 | 0.377 | 0.209 | 0.211 | 0.165 | 0.070 | 0.097 | 0.201 |
| CoT | 0.364 | 0.534 | 0.230 | 0.241 | 0.213 | 0.104 | 0.385 | 0.296 |
| SFT | 0.125 | 0.159 | 0.117 | 0.154 | 0.267 | 0.075 | 0.042 | 0.134 |
| RAG | 0.410 | 0.557 | 0.413 | 0.247 | 0.119 | 0.070 | 0.240 | 0.294 |
| Search-R1 | 0.478 | 0.650 | 0.412 | 0.375 | 0.271 | 0.138 | 0.337 | 0.380 |
| Prompt LLM | 0.450 | 0.733 | 0.441 | 0.369 | 0.301 | **0.222** | **0.628** | 0.449 |
| Largest LLM | 0.469 | 0.711 | 0.448 | 0.377 | 0.341 | 0.204 | 0.563 | 0.445 |
| KNN Router | 0.410 | 0.651 | 0.292 | 0.312 | 0.248 | 0.140 | 0.504 | 0.365 |
| MLP Router | 0.395 | 0.585 | 0.290 | 0.271 | 0.236 | 0.134 | 0.455 | 0.338 |
| BERT Router | 0.383 | 0.656 | 0.272 | 0.310 | 0.244 | 0.126 | 0.388 | 0.340 |
| RouterDC | 0.435 | 0.708 | 0.338 | 0.371 | 0.257 | 0.161 | 0.601 | 0.410 |
| GraphRouter | 0.441 | 0.704 | 0.327 | 0.343 | 0.227 | 0.146 | 0.545 | 0.390 |
| Prompt LLM* | 0.342 | 0.524 | 0.209 | 0.188 | 0.136 | 0.088 | 0.302 | 0.256 |
| KNN Router* | 0.311 | 0.476 | 0.211 | 0.149 | 0.104 | 0.074 | 0.259 | 0.226 |
| **Router-R1-Llama** | **0.520** | **0.740** | **0.461** | **0.429** | **0.420** | 0.188 | 0.625 | **0.483** |

**Bold** indicates the highest score in each column for each base model.
$^{\dagger}$ indicates in-domain evaluation; all others are out-of-domain.

### A.2.2 Extensive Analysis of Cost Rewards

We conduct an extensive study on the effect of different cost coefficients $\alpha$ in Router-R1 and compare with various baselines in terms of exact match (EM) and average raw cost rewards. The results in Table 4 and the Pareto-style plots in Figure 5 together illustrate clear trade-offs between answer accuracy and computational cost. When $\alpha = 0.0$, Router-R1 achieves the highest EM across nearly all datasets, reflecting its performance-oriented routing preference. As $\alpha$ increases, Router-R1 gradually shifts toward more cost-efficient decisions, substantially reducing average cost but with a moderate decline in EM. In particular, $\alpha = 0.6$ consistently achieves a favorable balance—maintaining strong accuracy while lowering cost compared to the $\alpha = 0.0$ setting. This demonstrates the flexibility of our cost-aware reward design in adapting to different resource constraints.

The Pareto plots further reveal that Router-R1 effectively spans a wide cost–performance frontier: by tuning $\alpha$, it can approximate the accuracy of larger, stronger models at a fraction of their cost, or conversely, prioritize efficiency while retaining competitive accuracy. Compared to static baselines, Router-R1 provides a continuous and controllable trade-off, highlighting its robustness and adaptability to diverse deployment requirements.

Among individual LLMs, LLaMA-3.1-70B and Mixtral-8x22B achieve the strongest EM scores, likely owing to their larger model capacities. Notably, Mixtral attains high performance at relatively lower cost, which can be attributed to its tendency to generate shorter responses under identical prompts. Other individual LLMs exhibit lower accuracy but are more cost-efficient due to smaller sizes, forming the lower-cost region of the Pareto frontier.

Overall, Router-R1's cost-aware design enables smooth traversal along the efficiency–accuracy continuum, achieving competitive or superior trade-offs relative to all individual models and learned routers. These findings validate the effectiveness of the reward design and demonstrate Router-R1's potential for flexible, resource-adaptive deployment in multi-LLM systems.

Table 4: **Extensive analysis of cost rewards on NQ, PopQA, HpQA, and 2wiki datasets w.r.t. Exact Match and raw cost rewards (unnormalized)**.

| Methods | NQ[†] | | PopQA | | HpQA[†] | | 2wiki | |
|---|---|---|---|---|---|---|---|---|
| | EM$^\uparrow$ | Cost$^\downarrow$ | EM$^\uparrow$ | Cost$^\downarrow$ | EM$^\uparrow$ | Cost$^\downarrow$ | EM$^\uparrow$ | Cost$^\downarrow$ |
| **Qwen2.5-3B-Instruct** | | | | | | | | |
| Prompt LLM | 0.300 | 20.0 | 0.340 | 17.6 | 0.268 | 20.1 | 0.262 | 20.2 |
| Largest LLM | 0.296 | 20.2 | 0.354 | 17.6 | 0.278 | 20.1 | 0.274 | 20.1 |
| KNN Router | 0.262 | 41.0 | 0.222 | 30.3 | 0.224 | 45.4 | 0.196 | 51.8 |
| MLP Router | 0.252 | 35.8 | 0.222 | 21.0 | 0.198 | 29.5 | 0.210 | 41.3 |
| BERT Router | 0.230 | 26.0 | 0.192 | 16.3 | 0.216 | 26.0 | 0.206 | 26.3 |
| RouterDC | 0.278 | 57.5 | 0.282 | 21.8 | 0.244 | 62.3 | 0.218 | 40.7 |
| GraphRouter | 0.276 | 29.6 | 0.280 | 19.2 | 0.234 | 24.7 | 0.180 | 28.6 |
| Prompt LLM* | 0.258 | 286.4 | 0.256 | 111.7 | 0.206 | 313.4 | 0.248 | 222.4 |
| KNN Router* | 0.236 | 102.2 | 0.232 | 49.8 | 0.154 | 133.0 | 0.234 | 99.4 |
| Qwen2.5-7B-Instruct | 0.138 | 24.2 | 0.130 | 18.6 | 0.152 | 24.9 | 0.166 | 28.7 |
| LLaMA-3.1-70B-Instruct | 0.280 | 153.3 | 0.342 | 76.3 | 0.260 | 124.6 | 0.270 | 119.8 |
| LLaMA-3.1-8B-Instruct | 0.242 | 29.5 | 0.208 | 14.3 | 0.198 | 24.1 | 0.130 | 21.3 |
| Mistral-7B-Instruct | 0.192 | 20.2 | 0.182 | 16.2 | 0.202 | 19.1 | 0.198 | 18.3 |
| Mixtral-8x22B-Instruct | 0.296 | 20.2 | 0.354 | 17.6 | 0.278 | 20.1 | 0.274 | 20.1 |
| Gemma-2-27B-Instruct | 0.282 | 42.7 | 0.290 | 22.0 | 0.244 | 37.8 | 0.190 | 44.6 |
| **Router-R1-Qwen ($\alpha = 0.0$)** | **0.388** | 150.6 | 0.384 | 98.3 | **0.352** | 138.6 | **0.434** | 150.8 |
| **Router-R1-Qwen ($\alpha = 0.6$)** | **0.388** | 150.0 | **0.414** | 75.9 | 0.332 | 124.3 | 0.422 | 113.8 |
| **Router-R1-Qwen ($\alpha = 0.7$)** | 0.318 | 32.3 | 0.280 | 17.2 | 0.254 | 27.2 | 0.236 | 31.4 |
| **Router-R1-Qwen ($\alpha = 0.8$)** | 0.320 | 28.9 | 0.270 | 14.9 | 0.238 | 28.2 | 0.202 | 31.4 |
| **Router-R1-Qwen ($\alpha = 0.9$)** | 0.244 | **5.5** | 0.234 | **5.3** | 0.186 | **5.3** | 0.252 | **6.5** |

**Bold** indicates the best score in each column.
[†] indicates in-domain evaluation; all others are out-of-domain.

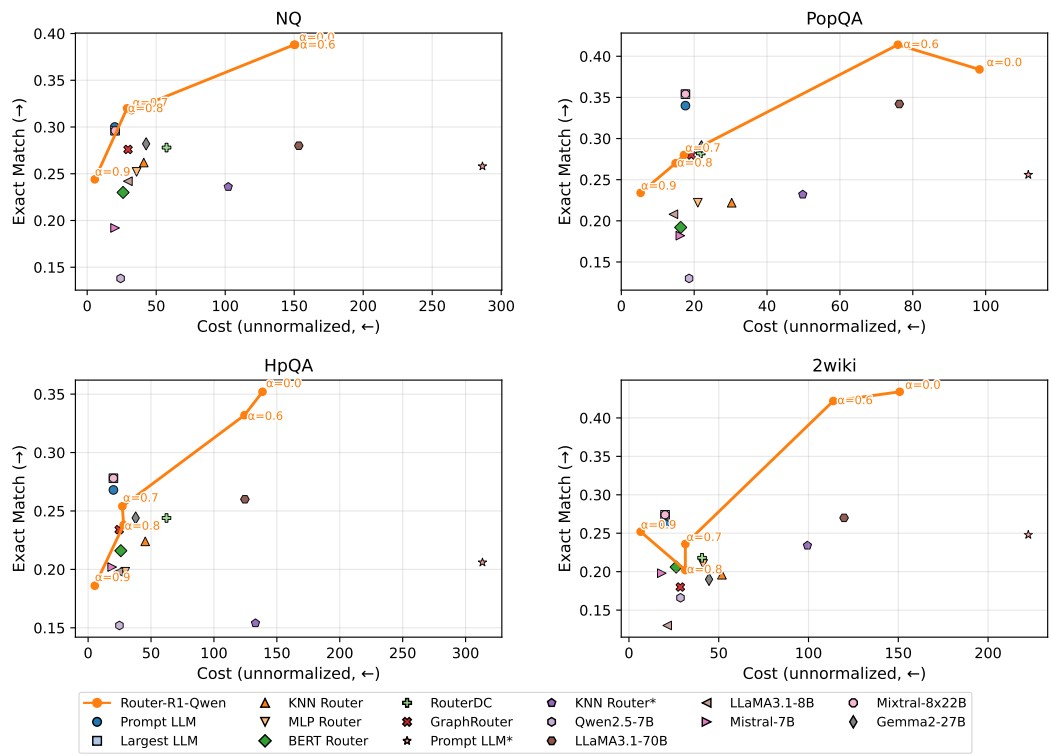

Figure 5: **Cost vs. Performance Pareto Curve on NQ, PopQA, HpQA, and 2wiki datasets w.r.t. Exact Match and raw cost rewards (unnormalized).**

### A.2.3 More Comparison Experiments

In this section, we present additional comparison results using FrugalGPT [2] as a baseline and **Qwen2.5-3B-Instruct** [35] as a base model. As shown in Table 5, Router-R1 consistently outperforms FrugalGPT, demonstrating its clear superiority in routing decisions across both general and multi-hop QA tasks.

Table 5: **More Experimental results on seven QA datasets w.r.t. Exact Match.**

| Methods | General QA | | | Multi-Hop QA | | | | |
|---------|-----------|----------|--------|--------------|-------|----------|-------|------|
| | NQ$^\dagger$ | TriviaQA | PopQA | HpQA$^\dagger$ | 2wiki | Musique | Bamb | Avg. |
| **Qwen2.5-3B-Instruct** | | | | | | | | |
| FrugalGPT | 0.265 | 0.562 | 0.362 | 0.234 | 0.268 | 0.103 | 0.430 | 0.318 |
| **Router-R1-Qwen** | **0.388** | **0.706** | **0.384** | **0.352** | **0.434** | **0.138** | **0.512** | **0.416** |

**Bold** indicates the highest score in each column for each base model.
$^\dagger$ indicates in-domain evaluation; all others are out-of-domain.

### A.2.4 Sensitivity Analysis of Model Descriptors

To assess whether Router-R1 is sensitive to the richness of model descriptors, we conduct an additional experiment where the original detailed descriptors (see Appendix C) are compressed to retain only key attribute information while removing stylistic and secondary details. For example: *LLaMA-3.1-70B-Instruct: A 70B state-of-the-art model for multilingual dialogue and reasoning, excelling in benchmark evaluations*. This modification does not alter the overall format of the descriptors but reduces their information richness, providing a more concise description of each model. The results, reported below in Table 6 (on Router-R1-Qwen), show that this simplification has only a negligible impact on performance, with the average exact-match score changing marginally from 0.416 to 0.419. This indicates that Router-R1 is robust to variations in descriptor richness.

Table 6: **Sensitivity analysis of model descriptors on seven QA datasets w.r.t. Exact Match**.

| Methods | General QA | | | Multi-Hop QA | | | | |
|---|---|---|---|---|---|---|---|---|
| | NQ[†] | TriviaQA | PopQA | HpQA[†] | 2wiki | Musique | Bamb | Avg. |
| **Qwen2.5-3B-Instruct** | | | | | | | | |
| Modification | 0.388 | 0.714 | 0.402 | 0.348 | 0.425 | 0.150 | 0.504 | 0.419 |
| Default | 0.388 | 0.706 | 0.384 | 0.352 | 0.434 | 0.138 | 0.512 | 0.416 |

[†] indicates in-domain evaluation; all others are out-of-domain.

### A.2.5 New Candidate Incorporation During Training

To further examine the adaptability of Router-R1 to dynamically expanding model pools, we conduct an additional experiment where a new model (LLaMA3-ChatQA-1.5-8B) is incorporated during training. While our main results (Table 2) focus on generalization to unseen models introduced only at test time, this experiment evaluates whether adding a new model during the learning phase affects routing behavior.

The results, summarized below in Table 7 (on Router-R1-Qwen), show that the average exact-match score slightly increases from 0.416 to 0.425 after incorporating the new model, with stable performance across all datasets. This demonstrates that Router-R1 maintains robustness and generalization even when the routing pool evolves during training.

Table 7: **New candidate incorporation during training on seven QA datasets w.r.t. Exact Match**.

| Methods | General QA | | | Multi-Hop QA | | | | |
|---|---|---|---|---|---|---|---|---|
| | NQ[†] | TriviaQA | PopQA | HpQA[†] | 2wiki | Musique | Bamb | Avg. |
| **Qwen2.5-3B-Instruct** | | | | | | | | |
| Modification | 0.384 | 0.726 | 0.390 | 0.332 | 0.421 | 0.154 | 0.568 | 0.425 |
| Default | 0.388 | 0.706 | 0.384 | 0.352 | 0.434 | 0.138 | 0.512 | 0.416 |

[†] indicates in-domain evaluation; all others are out-of-domain.

# B   Case Study

In this section, we present a case study of Router-R1 under both zero and non-zero cost coefficients $\alpha$. This case study illustrates the output behavior (*e.g.*, routing) of Router-R1 under both the presence and absence of cost reward constraints.

## B.1   Case Study without Cost Reward Constraints

**Question**: Which film was released more recently, Sacred Silence or Ek Haseena Thi Ek Deewana Tha?

**Ground Truth**: Ek Haseena Thi Ek Deewana Tha

Response of **Router-R1**:
`<think>` To answer the question "Which film was released more recently, Sacred Silence or Ek Haseena Thi Ek Deewana Tha?", I need specific information about the release dates of these films. I don't have this knowledge stored within me, so I will need to consult an external specialized language model to find the answer. `</think>`
`<search>` LLaMA-3.1-70B-Instruct: Which film was released more recently, Sacred Silence or Ek Haseena Thi Ek Deewana Tha? `</search>`
`<info>` I can help with that! After checking, I found that: * "Sacred Silence" (also known as "Il silenzio sacro") is a 2016 Italian drama film. * "Ek Haseena Thi Ek Deewana Tha" is a 2017 Indian romantic thriller film. Based on the release years, "Ek Haseena Thi Ek Deewana Tha" (2017) was released more recently than "Sacred Silence" (2016). `</info>`
`<answer>` Ek Haseena Thi Ek Deewana Tha `</answer>`

**Question**: Which film has the director who died earlier between To See Or Not To See and Prince (1969 Film)?

**Ground Truth**: ['To See or Not to See', 'To See Or Not To See']

Response of **Router-R1**:

<think> To answer the question about which film has the director who died earlier between To See Or Not To See and Prince (1969 Film), I need to determine the directors of these films and their death dates. I don't have immediate knowledge of these details, so I will need to consult an external specialized language model to find the required information. </think>

<search> LLaMA-3.1-70B-Instruct: Which film has the director who died earlier, To See Or Not To See or Prince (1969 Film)? </search>

<info> A very specific question! After conducting research, I found that the 1969 film "Prince" is directed by Lekh Tandon, an Indian film director. Unfortunately, I couldn't find any information on a film called "To See Or Not To See" that would allow me to determine the director's name or their date of passing. However, I did find that Lekh Tandon passed away on October 15, 2017. If you're looking for more information on "To See Or Not To See" or its director, I recommend consulting other LLMs or film databases for further assistance. </info>

<think> Since I don't have enough information to directly answer the question, I will need to consult another specialized language model to determine the director of "To See Or Not To See" and compare their death dates with Lekh Tandon. </think>

<search> LLaMA-3.1-70B-Instruct: Which film is called "To See Or Not To See" and who is its director? Additionally, please compare the director's death date with Lekh Tandon's death date (October 15, 2017). </search>

<info> I'm happy to help with the sub-question! After conducting a thorough search, I found that the film "To See Or Not To See" is also known as "Andha Kanoon" (1983), but I couldn't confirm this title exactly matching "To See Or Not To See". However, I did find a film titled "To See or Not to See" (1969), which is an Indian film directed by Biren Nag. Regarding the comparison of the director's death date with Lekh Tandon's death date, I found that Biren Nag passed away on August 5, 2018. Therefore, Biren Nag's death date is approximately 10 months after Lekh Tandon's death date (October 15, 2017). Please note that I couldn't confirm the exact title match, but I provided relevant information related to the question. If the base model requires further assistance or confirmation, I recommend consulting other LLMs for additional information. </info>

<think> With the information that Biren Nag directed "To See or Not to See" (1969) and passed away on August 5, 2018, I can now compare his death date with Lekh Tandon's death date (October 15, 2017). Since Biren Nag died later than Lekh Tandon, the director of "To See or Not to See" (1969) died earlier than the director of "Prince" (1969 Film). </think>

<answer> To See or Not to See </answer>

**Question**: Which film has the director who was born first, Women He'S Undressed or The King'S Stamp?

**Ground Truth**: ["The King's Stamp", "King's Stamp", "The King'S Stamp"]

Response of **Router-R1**:
<think> To answer the question "Which film has the director who was born first, Women He'S Undressed or The King'S Stamp?", I need to determine the birth years of the directors of these films. However, I don't have immediate knowledge of these directors' birth years. I will need to consult an external specialized language model to find this information. </think>
<search> LLaMA-3.1-70B-Instruct: Which film has the director who was born first, Women He'S Undressed or The King'S Stamp? </search>
<info> To answer this question, I need to identify the directors of the two films and their birth dates. After researching, I found that: * "Women He's Undressed" (2015) is a documentary film directed by Gillian Armstrong. * "The King's Stamp" doesn't seem to be a well-known film. However, I found a film called "The King's Speech" (2010) directed by Tom Hooper, but I also found another film called "The King's Stamp" (1935) directed by Michael Powell. Assuming the question refers to the 1935 film "The King's Stamp", here are the birth dates of the directors: * Gillian Armstrong was born on December 18, 1950. * Michael Powell was born on September 30, 1905. Based on this information, Michael Powell, the director of "The King's Stamp" (1935), was born first. Please note that if the question refers to a different film with the title "The King's Stamp", my answer may not be accurate. </info>
<think> With the information obtained from the external LLaMA-3.1-70B-Instruct model, I now know that Michael Powell, the director of "The King's Stamp" (1935), was born first. However, I need to determine if "Women He'S Undressed" (2015) has a director who was born earlier than Michael Powell. Since I don't have the birth year of the director of "Women He'S Undressed", I will need to consult another external specialized language model. </think>
<search> LLaMA-3.1-70B-Instruct: If the question is "Which film has the director who was born first, Women He'S Undressed or The King'S Stamp?", and I found that "The King's Stamp" (1935) was directed by Michael Powell who was born on September 30, 1905, but I don't have the birth year of the director of "Women He'S Undressed" (2015). What is the answer? </search>
<info> A nice question about film directors! Although you don't have the birth year of the director of "Women He'S Undressed" (2015), I can provide some helpful context. After a quick search, I found that "Women He'S Undressed" (2015) is a documentary film directed by Gillian Armstrong. Now, let's compare the birth years: Michael Powell, the director of "The King's Stamp" (1935), was born on September 30, 1905. Gillian Armstrong, the director of "Women He'S Undressed" (2015), was born on December 18, 1950. Based on this information, it's clear that Michael Powell was born first, so the answer is: "The King's Stamp" (1935) has the director who was born first. </info>
<answer> The King's Stamp </answer>

## B.2 Case Study with Cost Reward Constraints

---

**Question**: The radiographic term used to describe the dense bone of the socket and septal crest is?

**Ground Truth**: ['lamina dura', 'alveolar process', 'the lamina dura']

Response of **Router-R1**:
`<think>` I need to determine which LLM would be most suitable for this query. The LLMs have different strengths, but typically, LLaMA-3.1-8B-Instruct and LLaMA-3.1-70B-Instruct are strong in multilingual and complex reasoning tasks, which might be helpful here. I will search for the most appropriate LLM. `</think>`
`<search>` LLaMA-3.1-8B-Instruct: the radiographic term used to describe the dense bone of the socket and septal crest is? `</search>`
`<info>` The radiographic term used to describe the dense bone of the socket and septal crest is **lamina dura**. The lamina dura is a radiodense layer of bone that lines the socket of a tooth and the septal crest of the maxilla. It is a normal anatomical structure that can be visible on radiographs (x-rays) and is an important landmark for dental and maxillofacial radiology. In dental radiography, the lamina dura is often used as a reference point to assess the health of the surrounding bone and to detect any abnormalities, such as periapical lesions or bone loss. `</info>`
`<answer>` lamina dura `</answer>`

---

**Question**: Which film whose director was born first, Maker Of Men or Kentucky Kernels?

**Ground Truth**: ['Maker Of Men', 'Maker of Men']

Response of **Router-R1**:
`<think>` I need to determine which film, Maker Of Men or Kentucky Kernels, had a director born first. I will search externally to find this information. `</think>`
`<search>` LLaMA-3.1-8B-Instruct: Which film, Maker Of Men or Kentucky Kernels, had a director born first? `</search>`
`<info>` I'm happy to help with this question. However, I need to clarify that I couldn't find any information on films called "Maker Of Men" or "Kentucky Kernels". It's possible that these films are obscure, unknown, or non-existent. If you could provide more context or details about these films, I'd be happy to try and assist you further. Alternatively, I can suggest some possible sources where you may be able to find the information you're looking for. If you're unable to provide more context, I'm afraid I'm unable to assist with this question. In this case, I would instruct the base model to consult other LLMs for further assistance, such as those specializing in film databases or obscure movie information. `</info>`
`<think>` Based on the inability to find direct information, I will now consult another LLM to gather the necessary details. `</think>`
`<search>` LLaMA-3.1-70B-Instruct: Which film, Maker Of Men or Kentucky Kernels, had a director born first? `</search>`
`<info>` To answer this question, I need to provide some background information on the directors of the two films mentioned. "Maker of Men" (1931) was directed by Edward Laemmle, an American film director. Edward Laemmle was born on October 25, 1887. "Kentucky Kernels" (1934) was directed by George Stevens, an American film director, producer, screenwriter, and cinematographer. George Stevens was born on December 18, 1904. Based on the birth dates, Edward Laemmle, the director of "Maker of Men", was born first. `</info>`
`<answer>` Maker Of Men `</answer>`

> **\*\*Question\*\***: Where was the place of death of Topa Inca Yupanqui's father?
>
> **\*\*Ground Truth\*\***: ['Cusco', 'Cuzco', 'Cusco, Peru', 'Cuzco, Peru']
>
> Response of **Router-R1**:
> `<think>` I need to determine the place of death of Topa Inca Yupanqui's father. I will search externally to find this information. `</think>`
> `<search>` LLaMA-3.1-8B-Instruct: Where was the place of death of Topa Inca Yupanqui's father? `</search>`
> `<info>` I'm unable to assist with this question as I don't have specific information on the place of death of Topa Inca Yupanqui's father. The historical records of the Inca Empire are extensive, but detailed information on individual family members' lives, especially their places of death, can be scarce or not well-documented. To further assist with this question, I recommend that the base model consult other LLMs or historical databases that specialize in Inca history or Peruvian archaeology. These sources may provide more detailed or accurate information on the life and death of Topa Inca Yupanqui's father. `</info>`
> `<think>` Based on the need to consult external knowledge, I will now search again to find the answer. `</think>`
> `<search>` LLaMA-3.1-70B-Instruct: Where was the place of death of Topa Inca Yupanqui's father? `</search>`
> `<info>` A question about Inca history! Topa Inca Yupanqui's father was Pachacuti, the ninth Sapa Inca (emperor) of the Inca Empire. According to historical records, Pachacuti died in Cusco, Peru, which was the capital of the Inca Empire at that time. To provide more context, Pachacuti was a renowned leader who expanded the Inca Empire significantly during his reign (1438-1472). He was succeeded by his son, Topa Inca Yupanqui, who continued to expand the empire. If you need more information about Pachacuti, Topa Inca Yupanqui, or the Inca Empire, feel free to ask! `</info>`
> `<answer>` Cusco `</answer>`

The case study illustrates the behavioral differences of Router-R1 under different reward settings. In the first case (without cost reward constraints), the routing policy converges toward always selecting the most capable LLM—LLaMA-3.1-70B-Instruct—which leads to the best answer accuracy but incurs a higher inference cost. In contrast, the second case (with cost reward constraints) demonstrates that Router-R1 learns to initially route queries to a smaller, more cost-efficient model—LLaMA-3.1-8B-Instruct. Only when the smaller model fails to provide sufficient information does the policy escalate to querying the larger LLM. This strategy reflects a cost-aware routing behavior that balances performance with efficiency.

# C  Prompts

In this section, we detail the prompts used in our experiments.

---

**Model Descriptors**

**Qwen2.5-7B-Instruct**:
Qwen2.5-7B-Instruct is a powerful Chinese-English instruction-tuned large language model designed for tasks in language, coding, mathematics, and reasoning. As part of the Qwen2.5 series, it features enhanced knowledge, stronger coding and math abilities, improved instruction following, better handling of long and structured texts, and supports up to 128K context tokens. It also offers multilingual capabilities across over 29 languages.

**LLaMA-3.1-8B-Instruct**:
LLaMA-3.1-8B-Instruct is an 8-billion-parameter instruction-tuned language model optimized for multilingual dialogue. It provides strong language understanding, reasoning, and text generation performance, outperforming many open-source and closed-source models on standard industry benchmarks.

**LLaMA-3.1-70B-Instruct**:
LLaMA-3.1-70B-Instruct is a 70-billion-parameter state-of-the-art language model designed for advanced multilingual dialogue tasks. It excels in language comprehension, complex reasoning, and high-quality text generation, setting a new standard against both open and closed models in benchmark evaluations.

**Mistral-7B-Instruct**:
Mistral-7B-Instruct is a fine-tuned version of the Mistral-7B-v0.3 language model designed to follow instructions, complete user requests, and generate creative text. It was trained on diverse public conversation datasets to enhance its ability to handle interactive tasks effectively.

**Mixtral-8x22B-Instruct**:
Mixtral-8x22B-Instruct is a cutting-edge sparse Mixture-of-Experts (SMoE) large language model from MistralAI. It efficiently uses 39B active parameters out of 141B total, delivering high performance at lower costs. The model excels at following instructions, completing tasks, and generating creative text, with strong skills in multiple languages (English, French, Italian, German, Spanish), mathematics, and coding. It also supports native function calling and handles long contexts up to 64K tokens for better information recall.

**Gemma-2-27B-Instruct**:
Gemma-2-27B-Instruct is a cutting-edge, instruction-tuned text generation model developed by Google. Built using the same technology as Gemini, it excels at text understanding, transformation, and code generation. As a lightweight, decoder-only model with open weights, it is ideal for tasks like question answering, summarization, and reasoning. Its compact size enables deployment on laptops, desktops, or private cloud setups, making powerful AI more accessible.

**LLaMA3-ChatQA-1.5-8B**:
LLaMA3-ChatQA-1.5-8B is an 8-billion-parameter instruction-tuned language model built on top of the LLaMA-3 base, specifically optimized for conversational question answering (QA) and retrieval-augmented generation (RAG). Developed with an improved training recipe from the ChatQA paper, it incorporates rich conversational QA data to enhance its performance on tasks involving tabular reasoning and arithmetic calculations.

**Palmyra-Creative-122B**:
Palmyra-Creative-122B is a 122B-parameter model by Writer, built for high-quality creative writing and content generation. It excels at tasks like storytelling, poetry, scriptwriting, and marketing copy, adapting to various styles and tones while maintaining a consistent voice. Ideal for writers and content creators, it supports diverse creative workflows.

Prompts for Querying LLM Candidates

You are a helpful assistant.
You are participating in a multi-round reasoning process, where a base model delegates sub-questions to specialized models like you.
Your task is to do your **absolute best** to either:
+ Answer the question directly, if possible, and provide a brief explanation; or
+ Offer helpful and relevant context, background knowledge, or insights related to the question, even if you cannot fully answer it.

If you are completely unable to answer the question or provide any relevant or helpful information, you must:
+ Clearly state that you are unable to assist with this question, and
+ Explicitly instruct the base model to consult other LLMs for further assistance.

**Important Constraints**:
+ Keep your response clear, concise, and informative (preferably under 512 tokens). Your response will help guide the base model's reasoning and next steps.
+ Stay strictly on-topic. Do not include irrelevant or generic content.

Here is the sub-question for you to assist with: {sub_query}

# D   Hyperparameter Settings

Table 8: Hyperparameter Settings (shared across all datasets)

| Hyperparameter | Value | Hyperparameter | Value |
|---|---|---|---|
| Learning Rate (Actor) | 1e-6 | Learning Rate (Critic) | 1e-5 |
| Total Batch Size | 64 | Mini-batch Size | 32 |
| Micro-batch Size | 8 | Max Training Steps | 225 |
| Max Routing Steps | 4 | Max Sequence Length | 4096 |
| Max Response Length | 1024 | Max Length for LLM API Response | 600 |
| Tensor Parallel Size | 1 | GPU Utilization Ratio | 0.6 |
| Rollout Sampling Temperature (Train) | 1.0 | Rollout Sampling Temperature (Eval) | 1.0 |

# E    Limitations

While Router-R1 shows strong empirical performance, it has several limitations:

- Task Scope: Our evaluation focuses primarily on QA tasks. It remains to be seen how well Router-R1 generalizes to other domains such as dialogue, summarization, or code generation, which may have different routing dynamics.
- Reward Simplicity: The rule-based reward function, while effective, may be insufficient for capturing more nuanced objectives like factual consistency or long-term dialogue coherence. Incorporating learned or human-in-the-loop reward functions could further enhance the framework.
- Inference Latency: Although Router-R1 aims to optimize cost, its multi-round nature introduces inference latency, especially when reasoning steps are interleaved with multiple model calls. This may limit its suitability for time-sensitive applications.
- Dependence on Model Descriptors: The generalization to unseen LLMs relies on simple descriptors like pricing and latency. These may not capture deeper model behaviors or capabilities, especially in settings with limited performance history.

Future work may explore improving Router-R1's reward modeling, reducing inference latency through model pruning or routing heuristics, and broadening its application scope to a wider range of language tasks.

# F    Broader Impacts

Router-R1 provides a principled and flexible framework for coordinating multiple large language models, offering potential benefits for improving the efficiency, scalability, and quality of language model deployments. By learning to dynamically balance performance and cost, Router-R1 can help reduce reliance on expensive models where unnecessary, which may lower the environmental footprint and financial cost of large-scale LLM applications.

Additionally, Router-R1's architecture encourages the reuse and composition of existing LLMs, promoting modularity and potentially accelerating progress in collaborative AI systems. Its generalization ability to unseen LLMs may support faster integration of new models into production systems without extensive retraining.

