# OpenReview forum: "Router-R1: Teaching LLMs Multi-Round Routing and Aggregation via Reinforcement Learning"
_NeurIPS.cc/2025/Conference — NeurIPS 2025 poster_

### Official Review · Reviewer_f8xo · 2025-06-22

**Clarity:** 3
**Significance:** 3
**Originality:** 3
**Rating:** 4
**Confidence:** 3

**Summary:**

This paper proposes Router-R1, a new way of routing LLMs in multi-turns. Their solution is based on RL, where the router is trained to maximize a combination of different rewards.

**Questions:**

How does Eq 5 mitigate reward hacking?

**Ethical Concerns:**

["NO or VERY MINOR ethics concerns only"]

**Final Justification:**

They authors provided a new set of results I asked. Even though I have mixed feelings about the full set of results, this work offers a new approach to routing, which can handle multi-turn interactions (which is novel). I will increase my rating to 4.

**Limitations:**

Yes.

**Quality:**

2

**Strengths And Weaknesses:**

Strengths:
- The approach seems novel in two different ways:
    1. It uses RL for routing;
    2. It enables multi-turn routing which has not been explored before.

Weaknesses:
- The main issue with the paper is the following. One main experimental results is missing (but could probably be easily built from other results in the paper: the cost vs performance Pareto curves (shown, for example, in the RouterBench paper you cited). Ideally, you would have a plot for each benchmark and have a curve for each routing strategy. From that, it will be much clearer how your method compare to others; from the current results, I can't tell which method is performing better. This is the main reason I am suggesting borderline reject.
- In Eq 4 T_out is assumed to be known which is a limitation. Maybe you can predict it? See for example this paper:

Somerstep, Seamus, et al. "CARROT: A Cost Aware Rate Optimal Router." arXiv preprint arXiv:2502.03261 (2025).

---

> ### Author Rebuttal · Authors · 2025-07-30
>
> Thanks for your valuable feedback and appreciation of the clarity, significance, and contribution to our work. We appreciate your insights and would like to further address the concerns you raised about our work.
>
> ## cost vs performance Pareto curves
>
> Thank you for pointing this out, and we fully agree that visualizing the cost–performance trade-off is valuable for understanding routing strategy behavior.
>
> **Router-R1 was originally designed to optimize for answer performance**, and thus our main results (Table 2) focus on the α = 0 setting, where cost is not explicitly penalized. **The cost-aware reward term was included to give Router-R1 the flexibility to balance cost and performance when needed**, which enhances its practical value, though it was not the main focus of our comparison experiments due to space constraints.
>
> Nevertheless, we have already conducted an extensive study varying the cost coefficient α, evaluating how Router-R1 performs across different efficiency constraints. While we are unfortunately unable to include figures during the rebuttal phase (per NeurIPS policy), we include below a representative table of EM vs raw cost reward (unnormalized) across α values and baselines:
>
> | **Methods**                | **NQ EM↑** | **NQ Cost↓** | **PopQA EM↑** | **PopQA Cost↓** | **HpQA EM↑** | **HpQA Cost↓** | **2wiki EM↑** | **2wiki Cost↓** |
> | -------------------------- | ----------- | ------------- | ------------- | --------------- | ------------- | --------------- | ------------- | --------------- |
> | **Qwen2.5-3B-Instruct**    |             |               |               |                 |               |                 |               |                 |
> | Prompt LLM                 | 0.300       | 20.0          | 0.340         | 17.6            | 0.268         | 20.1            | 0.262         | 20.2            |
> | Largest LLM                | 0.296       | 20.2          | 0.354         | 17.6            | 0.278         | 20.1            | 0.274         | 20.1            |
> | KNN Router                 | 0.262       | 41.0          | 0.222         | 30.3            | 0.224         | 45.4            | 0.196         | 51.8            |
> | MLP Router                 | 0.252       | 35.8          | 0.222         | 21.0            | 0.198         | 29.5            | 0.210         | 41.3            |
> | BERT Router                | 0.230       | 26.0          | 0.192         | 16.3            | 0.216         | 26.0            | 0.206         | 26.3            |
> | RouterDC                   | 0.278       | 57.5          | 0.282         | 21.8            | 0.244         | 62.3            | 0.218         | 40.7            |
> | GraphRouter                | 0.276       | 29.6          | 0.280         | 19.2            | 0.234         | 24.7            | 0.180         | 28.6            |
> | Prompt LLM\*               | 0.258       | 286.4         | 0.256         | 111.7           | 0.206         | 313.4           | 0.248         | 222.4           |
> | KNN Router\*               | 0.236       | 102.2         | 0.232         | 49.8            | 0.154         | 133.0           | 0.234         | 99.4            |
> | **Router-R1-Qwen (α=0.0)** | **0.384**   | 149.4         | 0.391         | 97.5            | **0.313**     | 130.4           | **0.393**     | 145.5           |
> | **Router-R1-Qwen (α=0.6)** | **0.384**   | 150.3         | **0.421**     | 75.1            | 0.305         | 123.7           | 0.382         | 113.4           |
> | **Router-R1-Qwen (α=0.7)** | 0.314       | 31.4          | 0.285         | 16.9            | 0.236         | 25.9            | 0.214         | 31.2            |
> | **Router-R1-Qwen (α=0.8)** | 0.317       | 28.7          | 0.275         | 14.7            | 0.221         | 28.5            | 0.183         | 30.1            |
> | **Router-R1-Qwen (α=0.9)** | 0.241       | **5.4**       | 0.238         | **5.3**         | 0.176         | **5.4**         | 0.228         | **6.3**         |
>
> From this table, we observe:
> - At α = 0.0, Router-R1 consistently achieves the highest EM across benchmarks, confirming its strong performance orientation.
> - As α increases, cost drops significantly while EM gradually decreases, showing a clear trade-off.
> - Compared to other baselines, Router-R1 achieves better performance when cost is lightly constrained (small α), and more efficient routing under tighter budgets (larger α).
> These trends echo the Pareto-like patterns seen in works like RouterBench, and validate the strength and flexibility of our cost-aware reward design. We will include full cost–performance plots and analysis in the camera-ready version.
>
>
> ## T_out is assumed to be known which is a limitation
>
> Thank you for the suggestion. We believe there may be a slight misunderstanding here.
>
> In contrast to approaches like CARROT that rely on explicit cost and performance prediction to guide routing, Router-R1 does not require predicting T_out in advance. Instead, we adopt a reinforcement learning framework where the policy LLM learns to select the most suitable candidate LLMs by observing the actual outcomes of its decisions.
>
> Specifically, during training, the policy LLM selects a candidate LLM, receives its response (whose token count T_out is observed at runtime, not predicted), and incorporates that response into its own context. The reward is computed based on the final answer accuracy and the actual token cost (T_out). This reward then guides policy learning via RL, without requiring any explicit modeling of future token usage.
>
> In other words, Router-R1 delegates cost-performance estimation to the policy LLM itself, which is already a powerful model capable of learning complex trade-offs through experience, rather than relying on heuristic or learned cost predictors.
>
> We greatly appreciate the reviewer’s suggestion. Integrating ideas from CARROT, such as lightweight cost prediction, to further improve training efficiency or cold-start performance could be a promising future direction, and we plan to explore this in follow-up work.
>
> ## How does Eq 5 mitigate reward hacking?
>
> Thank you for the helpful question.
>
> While Eq. 5 defines the overall reward structure, we would like to clarify that Router-R1 adopts a hierarchical reward mechanism that is not explicitly shown in Eq. 5 for brevity (**as noted in Line 167**). This design plays a key role in mitigating reward hacking and improving training stability (**described in Line 166–172**).
>
> Concretely, Router-R1 assigns strict priorities among the three reward components: format > outcome > cost. If the format reward is -1 (e.g., due to invalid or unparseable output), the outcome and cost rewards are ignored (set to 0); If the outcome is wrong (i.e., outcome reward = 0), then cost reward is only considered if the outcome is correct, otherwise the total reward only depends on the format reward.
>
> Consider a situation where the output has the correct format but produces an incorrect answer. Without hierarchical reward, the model could still receive a positive total reward by minimizing the number of output tokens (i.e., reducing total T_out) and maximizing the cost reward. Over time, this may encourage the model to learn a shortcut: always produce very short, low-cost but incorrect answers, as this strategy still yields non-zero reward. This is a classic case of reward hacking. With hierarchical reward, however, once the outcome is incorrect, the cost reward is completely disregarded, and the total reward remains zero regardless of how low the cost is. This effectively eliminates such degenerate behaviors and ensures that correctness is prioritized over cheapness.
>
> In our experiments, we found that this hierarchical reward structure significantly stabilizes training and effectively mitigates reward hacking, leading to better final performance and robustness. We appreciate the reviewer for pointing out this important detail.
>
> Thank you again for your constructive feedback, and we commit to enriching the content of the paper based on the above in the revised version. We look forward to further engaging in discussions to improve the quality and impact of our work.

---

> > ### Comment · Reviewer_f8xo · 2025-08-02
> >
> > **Pareto curves:** I appreciate the inclusion of the Pareto curves and results looks reasonable. Thank you. I’d recommend the authors to further include the individual LLMs in the analysis. Could you upload a second version of the table with the individual LLMs, please?
> >
> > **Clarification about cost prediction:** Thank you for your clarification.  Now I understand this part!
> >
> > **Reward hacking:** Thank you for the clarification; it makes sense.
> >
> > For now, I will keep my scores and wait for the authors if they could provide the full set of results with (at least some) of the individual competitive models.

---

> > > ### Author Response · Authors · 2025-08-03
> > > **Response to Reviewer f8xo**
> > >
> > > We sincerely thank the reviewer for the prompt and constructive follow-up.
> > >
> > > Following your suggestion, we have updated our results and included a full version of the table with all individual LLMs used in our routing pool.
> > > | **Methods**                | **NQ EM↑** | **NQ Cost↓** | **PopQA EM↑** | **PopQA Cost↓** | **HpQA EM↑** | **HpQA Cost↓** | **2wiki EM↑** | **2wiki Cost↓** |
> > > | -------------------------- | ----------- | ------------- | ------------- | --------------- | ------------- | --------------- | ------------- | --------------- |
> > > | **Qwen2.5-3B-Instruct**    |             |               |               |                 |               |                 |               |                 |
> > > | Prompt LLM                 | 0.300       | 20.0          | 0.340         | 17.6            | 0.268         | 20.1            | 0.262         | 20.2            |
> > > | KNN Router                 | 0.262       | 41.0          | 0.222         | 30.3            | 0.224         | 45.4            | 0.196         | 51.8            |
> > > | MLP Router                 | 0.252       | 35.8          | 0.222         | 21.0            | 0.198         | 29.5            | 0.210         | 41.3            |
> > > | BERT Router                | 0.230       | 26.0          | 0.192         | 16.3            | 0.216         | 26.0            | 0.206         | 26.3            |
> > > | RouterDC                   | 0.278       | 57.5          | 0.282         | 21.8            | 0.244         | 62.3            | 0.218         | 40.7            |
> > > | GraphRouter                | 0.276       | 29.6          | 0.280         | 19.2            | 0.234         | 24.7            | 0.180         | 28.6            |
> > > | Prompt LLM\*               | 0.258       | 286.4         | 0.256         | 111.7           | 0.206         | 313.4           | 0.248         | 222.4           |
> > > | KNN Router\*               | 0.236       | 102.2         | 0.232         | 49.8            | 0.154         | 133.0           | 0.234         | 99.4            |
> > > |     |             |               |               |                 |               |                 |               |                 |
> > > | qwen2.5-7b-instruct **(UPDATED)**                | 0.138       | 24.2          | 0.130         | 18.6            | 0.152         | 24.9           | 0.166         | 28.7            |
> > > | llama-3.1-70b-instruct **(UPDATED)**                | 0.280       | 153.3          | 0.342         | 76.3            | 0.260         | 124.6            | 0.270         | 119.8            |
> > > | llama-3.1-8b-instruct **(UPDATED)**                | 0.242       | 29.5          | 0.208         | 14.3            | 0.198         | 24.1            | 0.130         | 21.3            |
> > > | mistral-7b-instruct-v0.3 **(UPDATED)**                | 0.192       | 20.2          | 0.182         | 16.2            | 0.202         | 19.1            | 0.198         | 18.3            |
> > > | mixtral-8x22b-instruct-v0.1 **(UPDATED)**                | 0.296       | 20.2          | 0.354         | 17.6            | 0.278         | 20.1            | 0.274         | 20.1            |
> > > | gemma-2-27b-it **(UPDATED)**                | 0.282       | 42.7          | 0.290         | 22.0            | 0.244         | 37.8            | 0.190         | 44.6            |
> > > |     |             |               |               |                 |               |                 |               |                 |
> > > | **Router-R1-Qwen (α=0.0)** | **0.384**   | 149.4         | 0.391         | 97.5            | **0.313**     | 130.4           | **0.393**     | 145.5           |
> > > | **Router-R1-Qwen (α=0.6)** | **0.384**   | 150.3         | **0.421**     | 75.1            | 0.305         | 123.7           | 0.382         | 113.4           |
> > > | **Router-R1-Qwen (α=0.7)** | 0.314       | 31.4          | 0.285         | 16.9            | 0.236         | 25.9            | 0.214         | 31.2            |
> > > | **Router-R1-Qwen (α=0.8)** | 0.317       | 28.7          | 0.275         | 14.7            | 0.221         | 28.5            | 0.183         | 30.1            |
> > > | **Router-R1-Qwen (α=0.9)** | 0.241       | **5.4**       | 0.238         | **5.3**         | 0.176         | **5.4**         | 0.228         | **6.3**         |
> > >
> > > Among individual LLMs, LLaMA3.1-70B and Mixtral-8x22B perform best, likely due to their larger sizes. Notably, Mixtral achieves strong results at lower cost, as it tends to generate shorter outputs despite identical prompts. Other individual LLMs show weaker performance but are also more cost-efficient due to smaller model sizes.
> > >
> > > Compared to individual LLMs, Router-R1 provides a flexible mechanism for navigating the performance-cost trade-off. By tuning α, it can approach the performance of stronger models at lower cost or reduce cost further while maintaining competitive accuracy. This highlights its adaptability to different resource and performance requirements.
> > >
> > > Thanks again for your helpful feedback and quick response!

---

> > > > ### Comment · Reviewer_f8xo · 2025-08-04
> > > >
> > > > Thank you for uploading the complete table! I have mixed feelings about the full set of results: it seems that for very low cost or very high cost, your routing method works the best. However, for intermediate costs, Mixtral offers the best results; in theory you should be doing at least as well as individual models, correct? I think this is a point you should look and try to improve. Anyways, because this works offers a new approach to routing, which can handle multi-turn interactions (which is novel), I will increase my rating to 4.

---

> > > > > ### Author Response · Authors · 2025-08-04
> > > > > **Response to Reviewer f8xo**
> > > > >
> > > > > Thank you for the prompt response and careful review. We greatly appreciate your constructive feedback and for increasing the rating.
> > > > >
> > > > > We actually observed the phenomenon you pointed out as well. Specifically, Mixtral appears to perform surprisingly well at intermediate cost levels. Upon closer inspection, we found that Mixtral tends to generate significantly shorter outputs compared to other large candidate models. To better understand this, we analyzed the average output lengths (in word count) of the candidate LLMs on the NQ dataset (with consistent input prompts). The results are shown below:
> > > > >
> > > > > | LLM                                       | Avg. Word Count |
> > > > > | ----------------------------------------- | --------------- |
> > > > > | qwen/qwen2.5-7b-instruct                  | 30.22           |
> > > > > | meta/llama-3.1-70b-instruct               | 104.23          |
> > > > > | meta/llama-3.1-8b-instruct                | 147.56          |
> > > > > | mistralai/mistral-7b-instruct-v0.3        | 37.30           |
> > > > > | mistralai/mixtral-8x22b-instruct-v0.1 | 39.54       |
> > > > > | google/gemma-2-27b-it                     | 20.39           |
> > > > >
> > > > > As shown above, Mixtral produces much shorter responses compared to, for example, llama-3.1-70b-instruct. This shorter output length likely contributes to its lower cost. We suspect this behavior may be related to the internal architecture of Mixtral as an MoE (Mixture-of-Experts) model: such models incorporate an internal "router" that activates only a subset of parameters during inference. In addition, differences in pretraining data and training objectives may also lead to inherently shorter responses—for example, if Mixtral was exposed to less multi-step reasoning content, it might tend to produce more concise outputs even when prompted to "think step-by-step".
> > > > >
> > > > > While these factors are interesting and likely relevant, further exploration into the exact causes goes beyond the current scope of this work. Nonetheless, your observation highlights an important point, and we will continue investigating how to improve Router-R1 under such conditions.
> > > > >
> > > > > Thank you again for your insightful feedback, helpful suggestions, and your time in reviewing our work. we sincerely appreciate your support!

---

> > > > > > ### Comment · Reviewer_f8xo · 2025-08-06
> > > > > >
> > > > > > Thank you for providing this last analysis!

---

### Official Review · Reviewer_7Web · 2025-06-22

**Clarity:** 3
**Significance:** 2
**Originality:** 3
**Rating:** 3
**Confidence:** 5

**Summary:**

Router-R1 is a multi-round collaborative routing and aggregation framework for LLMs, designed to overcome the limitations of existing routing approaches that assign each query to a single model in a one-shot manner. It formulates multi-model collaboration as a sequential decision process and introduces a *think–route* alternating mechanism: the router itself is an LLM with reasoning capabilities, which iteratively alternates between internal reasoning (*Think*) and model invocation (*Route*) across multiple rounds to progressively construct the final answer. To enhance training efficiency, Router-R1 employs a lightweight regularized reward function composed of format rewards, answer correctness rewards, and model invocation cost penalties, effectively guiding the model to balance accuracy and computational cost.

**Questions:**

Please see Weakness.

**Ethical Concerns:**

["NO or VERY MINOR ethics concerns only"]

**Limitations:**

yes

**Paper Formatting Concerns:**

no concerns

**Quality:**

3

**Strengths And Weaknesses:**

**Strengths**

1. The introduction of the cost reward is interesting and aligns well with the motivation for routing methods.
2. Experiments are comprehensive, but multi-run repetition experiments to validate stability appear lacking.
3. The paper is well written and clearly presented.

**Weaknesses**

1. The LLM routing pool lacks some state-of-the-art models, such as GPT-4o and DeepSeek-v3.
2. Although NeurIPS does not require reviewers to check arXiv versions of papers, I found a version of this paper on arXiv. I would like the authors to explain the discrepancy between the experimental results in the two versions, where the results are very different. If the paper is accepted, which version’s experimental results will be used in the camera-ready submission? (This will not affect my evaluation of the paper.)
3. In Section 5.3, the authors demonstrate the effect of introducing new models at test time for Router-R1. I am curious how Router-R1 would perform if new models, such as LLaMA3-ChatQA-1.5-8B, were incorporated during training.
4. Router-R1 requires multiple calls to LLMs, but Figure 2 shows that the average LLM API call count is only slightly above 1.0. This is puzzling because it implies most queries require only one call. Is multi-round calling really necessary? Does the current dataset adequately demonstrate the effectiveness of the proposed method? Additionally, how does Router-R1’s runtime efficiency compare with single-round invocation models?
5. The novelty of the paper is limited (though the introduction of the cost reward is commendable). While Router-R1 is presented as a routing method, from a modeling perspective, it resembles tool learning. For example, the authors mention search r1. The main difference here is that the authors treat the models in the LLM routing pool as callable tools. Although the distinction between these two approaches is discussed in Section 2.2, I find it not fundamentally different. The authors may consider modifying search r1 to better fit the routing scenario as a new baseline.

---

> ### Author Rebuttal · Authors · 2025-07-31
>
> Thanks for your valuable feedback and appreciation of the quality, clarity, and contribution to our work. We appreciate your insights and would like to further address the concerns you raised about our work.
>
> ## The LLM routing pool lacks some state-of-the-art models
>
> We sincerely thank the reviewer for raising this point.
> Due to limited GPU resources and budget constraints, our experiments rely on public LLM APIs to access candidate models (Lines 257–263). We carefully selected a representative and diverse set of models. However, some cutting-edge models like GPT-4o are either unavailable via the APIs we use or incur prohibitive costs and latency, making them infeasible for repeated runs in academic settings.
>
> Moreover, Router-R1 is inherently model-agnostic—its routing policy can generalize to any future LLMs, including stronger ones, without requiring algorithmic changes. We hope our work will inspire industrial labs with greater compute resources to further evaluate Router-R1 at scale with larger routing pools.
>
> As for reproducibility, we ran multiple repetitions and observed consistently low variance (typically within ±0.01 EM), which does not affect the conclusions. Reported results are averaged across runs, and we omit variance values for clarity. We will also open-source our code upon publication to support reproducibility.
>
> ## The results in arXiv version
>
> Thanks for your feedback.
> To ensure compliance with NeurIPS's double-blind review policy and maintain fairness in the reviewing process, we have communicated the relevant details to the Area Chair.
> We appreciate your attention and understanding regarding this matter.
>
> ## New model incorporation
>
> Thank you for the suggestion. While we primarily evaluated Router-R1’s generalization to new models introduced at test time (Table 3), we also experimented with incorporating a new model (LLaMA3-ChatQA-1.5-8B) during training.
> Below we report the average EM scores across datasets when this model is added to the routing pool at training time (Router-R1-Qwen):
> | Setting         | NQ    | TriviaQA | PopQA | HotpotQA | 2wikimultihop | Musique | BAMB  | Avg   |
> |-----------------|-------|----------|-------|----------|----------------|---------|--------|--------|
> | Modification | 0.380 | 0.719    | 0.397 | 0.295    | 0.381          | 0.167   | 0.589  | 0.418 |
> | Default | 0.384 | 0.699    | 0.391 | 0.313    | 0.393          | 0.150   | 0.531  | 0.409 |
>
> This result aligns closely with the test-time-only setting (Table 3), suggesting that Router-R1 generalizes well to new models and remains robust even when the routing pool is dynamically extended.
>
> ## Multi-round Calling Frequency
>
> The number of LLM calls required per query depends on multiple factors, notably **(1) the complexity of the dataset**, and **(2) the composition of the routing pool**.
> To understand how task complexity affects routing depth, we conduct a quantitative analysis using the average EM performance across all candidate LLMs on each dataset. This gives a proxy of dataset difficulty (harder datasets typically have lower EM scores).
> | Dataset | NQ    | TriviaQA | PopQA | HotpotQA | 2Wiki | Musique | BAMB  |
> | ------- | ----- | -------- | ----- | -------- | ----- | ------- | ----- |
> | Avg EM  | 0.257 | 0.537    | 0.220 | 0.296    | 0.298 | 0.133   | 0.412 |
>
> This aligns with intuition (e.g., multi-hop datasets like Musique are harder), and when correlated with Figure 2a, it shows that more difficult datasets tend to invoke more LLM calls. This supports the motivation for our multi-round routing framework, which flexibly allocates computational resources based on query difficulty.
>
> Moreover, the routing pool itself affects the observed call frequency. Due to API constraints, we use a pool of API-accessible LLMs; stronger routing pools would naturally lead to fewer calls per query. **However, this does not undermine the necessity or effectiveness of multi-round routing, especially when combined with cost-aware rewards.** Importantly, multi-round routing enables behaviors that one-shot methods cannot learn. For example, with cost reward enabled, Router-R1 often learns to first query smaller models, escalating to larger ones only when needed (Appendix B.2). This adaptive, efficient strategy is not achievable with one-shot routing. Finally, while current experiments focus on QA tasks, we believe the general-purpose design of Router-R1 allows easy extension to more complex reasoning domains in future work.
>
> ## Runtime efficiency
>
> We acknowledge the importance of runtime efficiency. However, a direct comparison between Router-R1 and single-round baselines in terms of wall-clock runtime would be unfair, as our setup involves heterogeneous inference environments. Candidate LLMs are accessed via APIs (e.g., Together AI), which often apply advanced backend optimizations for fast inference. In contrast, Router-R1's policy model runs locally on a single GPU without such system-level optimizations.
> Instead, we report inference cost w.r.t. raw cost rewards, which fairly reflects efficiency and is commonly used in prior work. As shown in the following table, Router-R1 achieves a strong cost-performance trade-off under this metric.
> | **Methods**                | **NQ EM↑** | **NQ Cost↓** | **PopQA EM↑** | **PopQA Cost↓** | **HpQA EM↑** | **HpQA Cost↓** | **2wiki EM↑** | **2wiki Cost↓** |
> | -------------------------- | ----------- | ------------- | ------------- | --------------- | ------------- | --------------- | ------------- | --------------- |
> | **Qwen2.5-3B-Instruct**    |             |               |               |                 |               |                 |               |                 |
> | Prompt LLM                 | 0.300       | 20.0          | 0.340         | 17.6            | 0.268         | 20.1            | 0.262         | 20.2            |
> | Largest LLM                | 0.296       | 20.2          | 0.354         | 17.6            | 0.278         | 20.1            | 0.274         | 20.1            |
> | KNN Router                 | 0.262       | 41.0          | 0.222         | 30.3            | 0.224         | 45.4            | 0.196         | 51.8            |
> | MLP Router                 | 0.252       | 35.8          | 0.222         | 21.0            | 0.198         | 29.5            | 0.210         | 41.3            |
> | BERT Router                | 0.230       | 26.0          | 0.192         | 16.3            | 0.216         | 26.0            | 0.206         | 26.3            |
> | RouterDC                   | 0.278       | 57.5          | 0.282         | 21.8            | 0.244         | 62.3            | 0.218         | 40.7            |
> | GraphRouter                | 0.276       | 29.6          | 0.280         | 19.2            | 0.234         | 24.7            | 0.180         | 28.6            |
> | Prompt LLM\*               | 0.258       | 286.4         | 0.256         | 111.7           | 0.206         | 313.4           | 0.248         | 222.4           |
> | KNN Router\*               | 0.236       | 102.2         | 0.232         | 49.8            | 0.154         | 133.0           | 0.234         | 99.4            |
> | **Router-R1-Qwen (α=0.0)** | **0.384**   | 149.4         | 0.391         | 97.5            | **0.313**     | 130.4           | **0.393**     | 145.5           |
> | **Router-R1-Qwen (α=0.6)** | **0.384**   | 150.3         | **0.421**     | 75.1            | 0.305         | 123.7           | 0.382         | 113.4           |
> | **Router-R1-Qwen (α=0.7)** | 0.314       | 31.4          | 0.285         | 16.9            | 0.236         | 25.9            | 0.214         | 31.2            |
> | **Router-R1-Qwen (α=0.8)** | 0.317       | 28.7          | 0.275         | 14.7            | 0.221         | 28.5            | 0.183         | 30.1            |
> | **Router-R1-Qwen (α=0.9)** | 0.241       | **5.4**       | 0.238         | **5.3**         | 0.176         | **5.4**         | 0.228         | **6.3**         |
>
>
> ## Novelty
>
> Our work is motivated by the need for multi-round, adaptive routing in complex reasoning tasks—something one-shot routing fails to handle effectively. This requires a centralized policy LLM to coordinate step-wise model calls, where each candidate LLM functions as a specialized reasoning module rather than a generic tool. Inspired by DeepSeek-R1, we introduce reinforcement learning to train the policy LLM, enabling fine-grained and adaptive routing. To our knowledge, Router-R1 is the first to apply R1-style RL to LLM routing, unifying multi-round reasoning with cost-aware inference (**Lines 31–39**). This sets it apart from both tool learning paradigms and prior router-style works.
>
> A core innovation is our cost reward design, which allows Router-R1 to learn efficient, hierarchical routing behaviors—favoring small models and escalating only when needed (Appendix B.2). This dynamic trade-off between cost and performance is crucial in real-world, cost-sensitive deployments and cannot be achieved by fixed, one-shot methods.
>
> While Search-R1 also uses RL, it addresses tool use rather than model routing. Unlike tool selection among heterogeneous modules (e.g., calculators)[1,2,3], Router-R1 routes across semantically similar LLMs differing in capability and cost, which is a fundamentally different challenge. We also include Search-R1 as a baseline (Appendix A.2), but emphasize that Router-R1 is a standalone contribution tailored for multi-round LLM routing, not an extension of tool learning.
>
> [1] Toolformer: Language Models Can Teach Themselves to Use Tools. NeurIPS 2023.
>
> [2] CRITIC: Large Language Models Can Self-Correct with Tool-Interactive Critiquing. ICLR 2024.
>
> [3] ToolRL: Reward is All Tool Learning Needs. arXiv 2025.
>
> Thank you again for your constructive feedback, and we commit to enriching the content of the paper based on the above in the revised version. We look forward to further engaging in discussions to improve the quality and impact of our work.

---

> > ### Comment · Reviewer_7Web · 2025-08-04
> >
> > Thank you for the response. What is the AC's reply regarding the results in the arXiv version?

---

> > > ### Author Response · Authors · 2025-08-04
> > > **Response to Reviewer 7Web**
> > >
> > > Thank you again for the thoughtful follow-up. In some cases, discrepancies may occur between preliminary versions prepared for broader sharing and the official conference submission. These are often due to standard refinements such as minor hyperparameter adjustments or updates to improve performance and reproducibility, which do not affect the paper's core claims or conclusions.
> > >
> > > From what we understand, the area chairs have acknowledged that such differences are acceptable and not grounds for concern, provided they arise from typical post-deadline improvements and do not undermine the integrity of the submission. In such cases, the final version typically reflects the most refined and up-to-date results, consistent with the preliminary version intended for broader sharing.
> > >
> > > We sincerely apologize if any parts of our previous response were unclear. Due to the double-blind policy, we were cautious in our phrasing and limited in how much detail we could provide. We appreciate your understanding.

---

> > > > ### Comment · Reviewer_f8xo · 2025-08-04
> > > >
> > > > I think Reviewer 7Web has a point on evaluating SOTA. You could probably do it for free using datasets other works have provided, e.g., SPROUT (https://huggingface.co/datasets/CARROT-LLM-Routing/SPROUT). They have info about the cost and correctness of LLM responses and include frontier LLMs.

---

> > > > > ### Author Response · Authors · 2025-08-04
> > > > > **Response to Reviewer f8xo**
> > > > >
> > > > > Thank you for the suggestion and for pointing us to the SPROUT dataset.
> > > > >
> > > > > This is indeed a valuable and well-curated resource. As a static dataset that provides correctness and cost annotations for a variety of LLM responses, SPROUT is highly suitable for training LLM routers that rely on offline data, such as RouterDC and GraphRouter. It also includes several advanced and frontier LLMs such as GPT-4o, which is particularly valuable given its limited availability. Moreover, the use of LLM-based evaluation to assess candidate responses is a promising direction that goes beyond traditional automatic metrics, and one that could help improve router robustness in future research.
> > > > >
> > > > > However, Router-R1 differs substantially in its training paradigm. It is designed as an online reinforcement learning approach, where the policy LLM dynamically interacts with the routing pool in real time during training. This process involves routing at the sub-query level and actively collecting feedback during execution. As a result, the data distribution evolves during training, and the system benefits from exploration and exploitation trade-offs that are not available in pre-collected static datasets. The fact that Router-R1 delegates different sub-queries to different LLMs adds further complexity, making it challenging to directly reuse datasets like SPROUT for our setting.
> > > > >
> > > > > Overall, the experimental setting of Router-R1 is quite different from that of CARROT, especially in terms of training dynamics and data granularity. That being said, we truly appreciate the availability of SPROUT. In fact, we have also constructed a dataset in a similar format (see Lines 232–237), containing cost and performance metrics of several strong LLMs on our chosen QA benchmarks. This was used to support the training of baseline methods that require static supervision.
> > > > >
> > > > > We believe that datasets like SPROUT will play a key role in building shared benchmarks and reducing entry barriers for researchers with limited computational budgets, thereby advancing the LLM routing community. We are glad to be made aware of this resource and will be mindful of opportunities to make use of it and support the growth of this shared ecosystem in future work. We also plan to include a reference link to this resource on our upcoming open-source GitHub page to help foster a more connected and collaborative environment for LLM router research.
> > > > >
> > > > > Thank you again for your thoughtful suggestion and the time you’ve dedicated to reviewing our work!

---

> > > ### Author Response · Authors · 2025-08-07
> > > **Response to Reviewer 7Web**
> > >
> > > Dear Reviewer 7Web,
> > >
> > > We understand that the discussion phase may be a busy time, and we sincerely appreciate the thoughtful engagement you've shown so far.
> > >
> > > As the deadline approaches, we would be grateful to hear any further thoughts you might have. We hope our previous responses and additional experiments have helped clarify the points you raised. If you feel that your concerns have been adequately addressed, we would kindly ask you to consider updating your review score accordingly.
> > >
> > > Your feedback is very important to us, and we truly value the time and effort you've invested in the review process.
> > >
> > > Thank you again!

---

> ### Author Response · Authors · 2025-08-09
> **Response to Reviewer 7Web**
>
> Dear Reviewer 7Web,
>
> With only a few hours left in the discussion phase, we would greatly appreciate it if you could share any final thoughts or consider updating your score if our responses have addressed your concerns.
>
> Thank you again for your time and effort.

---

### Official Review · Reviewer_Gukk · 2025-07-02

**Clarity:** 4
**Significance:** 3
**Originality:** 3
**Rating:** 4
**Confidence:** 3

**Summary:**

The paper tackles the limitation of current LLM routers that make a single one-shot model choice. It formulates routing as a sequential decision process in which a router-LLM alternates between internal reasoning,  external model calls and final answering. The router is trained end-to-end with PPO on a composite reward (format, exact-match accuracy, and cost). Experiments on seven QA benchmarks show sizable gains over strong one-shot routers and retrieval baselines, while a cost-coefficient study demonstrates controllable performance/price trade-offs.

**Questions:**

1. How sensitive is performance to the number or quality of model descriptors supplied in the prompt?

2. Will the approach scales to larger policies with parameter-efficient tuning like LORA?

**Ethical Concerns:**

["NO or VERY MINOR ethics concerns only"]

**Limitations:**

Yes.

**Paper Formatting Concerns:**

No concerns

**Quality:**

3

**Strengths And Weaknesses:**

Strength
- Clear formulation of multi-round routing as RL.
- Lightweight yet effective reward design.
- Strong empirical results demonstrate generalization beyond in-domain datasets.

Weaknesses
- Not consider multi-round Router cost in comparison. The cost of policy router in Router-R1 can dominate latency when the called model is a small model.  Other baselines perform a single forward pass, omitting the router’s share makes Router-R1’s cost look lower than it really is.
- Missing baselines. The comparison set lacks other multi-step methods. For example: Lingjiao Chen, Matei Zaharia, James Zou. FrugalGPT: How to Use Large Language Models While Reducing Cost and Improving Performance.
- Router scalability. The router is a 3 B-parameter model trained with full-parameter PPO,  it is unclear whether the approach scales to larger policies with parameter-efficient tuning.

---

> ### Author Rebuttal · Authors · 2025-07-30
>
> Thanks for your valuable feedback and appreciation of the quality, clarity, and contribution to our work. We appreciate your insights and would like to further address the concerns you raised about our work.
>
> ## Inference Costs
>
> We thank the reviewer for raising this important point and would like to offer some clarifications and insights regarding the cost evaluation of Router-R1.
>
> > (1) Goal of the main experiment: performance-oriented comparison
>
> In the main results table (Table 2), we set the cost reward coefficient $\alpha = 0.0$ to evaluate the performance ceiling of Router-R1, focusing on its multi-round reasoning capabilities. The primary goal here was not cost-efficiency but answer quality under best-effort routing. Due to space limitations, cost-performance tradeoffs were deferred to a separate analysis.
>
> > (2) Cost-performance tradeoff analysis
>
> We conduct a comprehensive study on different cost coefficients $\alpha$ in Router-R1, comparing with baselines in terms of exact match (EM) and raw cost rewards (unnormalized).
> As shown in the following table, with $\alpha = 0.0$, Router-R1 achieves the highest EM, demonstrating its effectiveness in performance-first scenarios. As $\alpha$ increases, the total routing cost drops significantly, while EM degrades moderately—highlighting the controllable trade-off between accuracy and efficiency.
> | **Methods**                | **NQ EM↑** | **NQ Cost↓** | **PopQA EM↑** | **PopQA Cost↓** | **HpQA EM↑** | **HpQA Cost↓** | **2wiki EM↑** | **2wiki Cost↓** |
> | -------------------------- | ----------- | ------------- | ------------- | --------------- | ------------- | --------------- | ------------- | --------------- |
> | **Qwen2.5-3B-Instruct**    |             |               |               |                 |               |                 |               |                 |
> | Prompt LLM                 | 0.300       | 20.0          | 0.340         | 17.6            | 0.268         | 20.1            | 0.262         | 20.2            |
> | Largest LLM                | 0.296       | 20.2          | 0.354         | 17.6            | 0.278         | 20.1            | 0.274         | 20.1            |
> | KNN Router                 | 0.262       | 41.0          | 0.222         | 30.3            | 0.224         | 45.4            | 0.196         | 51.8            |
> | MLP Router                 | 0.252       | 35.8          | 0.222         | 21.0            | 0.198         | 29.5            | 0.210         | 41.3            |
> | BERT Router                | 0.230       | 26.0          | 0.192         | 16.3            | 0.216         | 26.0            | 0.206         | 26.3            |
> | RouterDC                   | 0.278       | 57.5          | 0.282         | 21.8            | 0.244         | 62.3            | 0.218         | 40.7            |
> | GraphRouter                | 0.276       | 29.6          | 0.280         | 19.2            | 0.234         | 24.7            | 0.180         | 28.6            |
> | Prompt LLM\*               | 0.258       | 286.4         | 0.256         | 111.7           | 0.206         | 313.4           | 0.248         | 222.4           |
> | KNN Router\*               | 0.236       | 102.2         | 0.232         | 49.8            | 0.154         | 133.0           | 0.234         | 99.4            |
> | **Router-R1-Qwen (α=0.0)** | **0.384**   | 149.4         | 0.391         | 97.5            | **0.313**     | 130.4           | **0.393**     | 145.5           |
> | **Router-R1-Qwen (α=0.6)** | **0.384**   | 150.3         | **0.421**     | 75.1            | 0.305         | 123.7           | 0.382         | 113.4           |
> | **Router-R1-Qwen (α=0.7)** | 0.314       | 31.4          | 0.285         | 16.9            | 0.236         | 25.9            | 0.214         | 31.2            |
> | **Router-R1-Qwen (α=0.8)** | 0.317       | 28.7          | 0.275         | 14.7            | 0.221         | 28.5            | 0.183         | 30.1            |
> | **Router-R1-Qwen (α=0.9)** | 0.241       | **5.4**       | 0.238         | **5.3**         | 0.176         | **5.4**         | 0.228         | **6.3**         |
>
> We will include this expanded analysis in the camera-ready version for completeness.
>
> > (3) Ensuring fairness in comparison
>
> To ensure fairness when comparing with single-step methods, we introduced two multi-round routing baselines (Prompt LLM*, KNN Router*) that simulate step-by-step decomposition.
> Furthermore, we note that most prior LLM router works only include the cost of routed LLM calls and similarly omit the router's cost in their analysis[1][2].
> To ensure consistency and fair comparison, we follow this practice and focus our cost evaluation on the routed candidate LLMs, which dominate the total computational overhead.
>
> > (4) Policy LLM cost is minimal, and Router-R1 enables scalable offloading
>
> We acknowledge the concern about the policy LLM’s potential latency or cost, especially when routing to smaller models. However, in our current setup, the policy LLM is only 3B, significantly smaller than the candidate LLMs.
> Moreover, as shown in Appendix B (Page 23), the policy LLM performs only lightweight coordination tasks, and the vast majority of output tokens are generated by candidate LLMs.
> In fact, Router-R1 can reduce latency by delegating parts of the reasoning process to smaller, faster models in the routing pool, instead of relying solely on the policy model.
> This architecture is particularly beneficial when the policy LLM is larger than some candidate models (a setting increasingly common in practical deployments). Even though our current policy model is lightweight, the Router-R1 framework naturally supports scalable offloading, where a central controller coordinates efficient model usage. This can even lead to lower latency and cost than using a single monolithic model, especially for simple tasks. Interestingly, this resembles ideas in speculative decoding[3], where small models generate drafts and large models verify them, reducing end-to-end inference time and cost.
>
> In summary, Router-R1 adopts a flexible and extensible architecture that balances reasoning quality and computational efficiency. We take care to ensure fair cost comparison, and we appreciate the reviewer’s thoughtful feedback, which we will further address in the final version.
>
> [1] GraphRouter: A Graph-based Router for LLM Selections. ICLR 2025.
>
> [2] RouterDC: Query-Based Router by Dual Contrastive Learning for Assembling Large Language Models. NeurIPS 2024.
>
> [3] Fast Inference from Transformers via Speculative Decoding. ICML 2023.
>
> ## Missing Baselines
>
> We thank the reviewer for pointing out this baseline.
> We did evaluate FrugalGPT in our experiments. However, we chose not to include it in the main results table, as its design prioritizes cost reduction, whereas Router-R1 is primarily aimed at maximizing performance under multi-round decomposition, especially in complex reasoning settings.
> We agree that including FrugalGPT helps contextualize our method across different optimization objectives. Below we provide its results for completeness:
> | Model           | NQ    | TriviaQA | PopQA | HotpotQA | 2wikimultihop | Musique | BAMB  | Avg   |
> |-------------------|-------|----------|-------|----------|----------------|---------|--------|--------|
> | FrugalGPT         | 0.265 | 0.562    | 0.362 | 0.234    | 0.268          | 0.103   | 0.430  | 0.318 |
> | Router-R1-Qwen    | 0.384 | 0.699    | 0.391 | 0.313    | 0.393          | 0.150   | 0.531  | 0.409 |
>
> We will include this comparison in the camera-ready version to better position our method within the broader landscape of routing strategies.
>
> ## Router scalability
>
> Thank you for raising this question.
> Our method is built upon the VERL framework, which already supports parameter-efficient tuning methods such as LoRA. This means that Router-R1 can be directly scaled to larger policy models using LoRA or similar approaches, without requiring full-parameter training.
> However, our focus in this work is not on optimizing parameter efficiency, but rather on proposing a multi-round, cost-aware LLM routing strategy. Exploring how Router-R1 behaves under different tuning paradigms (e.g., LoRA or QLoRA) is certainly interesting, and we consider it promising future work.
>
> ## Sensitivity of Model descriptors
>
> Thank you for the insightful question.
> To evaluate the sensitivity of model descriptors, we conducted an additional experiment where we compressed the original prompts (shown in Appendix B) to retain only basic attribute information. For example:
> > LLaMA-3.1-70B-Instruct: A 70B state-of-the-art model for multilingual dialogue and reasoning, excelling in benchmark evaluations.
>
> Compared to the more elaborate descriptors used in our default setup, this version is significantly more concise.
> The results are shown in the table below (on Router-R1-Qwen). We observe that such modification has minimal impact on overall performance, indicating that Router-R1 is not highly sensitive to the model descriptors.
> | Setting         | NQ    | TriviaQA | PopQA | HotpotQA | 2wikimultihop | Musique | BAMB  | Avg   |
> |-----------------|-------|----------|-------|----------|----------------|---------|--------|--------|
> | Modification  | 0.384 | 0.707    | 0.409 | 0.309    | 0.385          | 0.163   | 0.523  | 0.411 |
> | Default | 0.384 | 0.699    | 0.391 | 0.313    | 0.393          | 0.150   | 0.531  | 0.409 |
>
> As discussed in the paper (**Line 188–195**), model descriptors primarily serve as a cold-start prior, helping the router bootstrap the learning process. As training progresses, Router-R1 learns to distinguish the strengths and weaknesses of candidate LLMs through interaction, rather than relying solely on initial descriptors.
> We will include this result in the camera-ready version.
>
>
> Thank you again for your constructive feedback, and we look forward to further engaging in discussions to improve the quality and impact of our work.

---

> ### Author Response · Authors · 2025-08-07
> **Response to Reviewer Gukk**
>
> Dear Reviewer Gukk,
>
> We understand that the discussion phase may be a busy time, and we sincerely appreciate the thoughtful engagement you've shown so far.
>
> As the deadline approaches, we would be grateful to hear any further thoughts you might have. We hope our previous responses and additional experiments have helped clarify the points you raised. If you feel that your concerns have been adequately addressed, we would kindly ask you to consider updating your review score accordingly.
>
> Your feedback is very important to us, and we truly value the time and effort you've invested in the review process.
>
> Thank you again!

---

> ### Author Response · Authors · 2025-08-09
> **Response to Reviewer Gukk**
>
> Dear Reviewer Gukk,
>
> With only a few hours left in the discussion phase, we would greatly appreciate it if you could share any final thoughts or consider updating your score if our responses have addressed your concerns.
>
> Thank you again for your time and effort.

---

### Official Review · Reviewer_iXcu · 2025-07-05

**Clarity:** 4
**Significance:** 2
**Originality:** 3
**Rating:** 5
**Confidence:** 4

**Summary:**

This paper introduces an interesting idea to combine model reasoning ability with multi-model routing,  Router-R1 optimizes for performance and cost using a hierarchal rule-based reward system. It demonstrates strong generalization to unseen LLMs via simple model descriptors. Experiments show superior performance on diverse QA benchmarks, highlighting robust generalization and cost management.

**Questions:**

- is it possible to consider the computational cost of the Router-R1 policy LLM into the overall cost analysis, or even let the model optimize the joint cost?

**Ethical Concerns:**

["NO or VERY MINOR ethics concerns only"]

**Final Justification:**

My original recommendation of this paper is positive. While I and some reviewers are concerned about the cost of the router, I consider that is a factor that might change as technology develops so I wouldn't use that as the main reason to reject this work.

My recommendation is in between borderline accept/accept, but in general I think the idea is novel and I try not to use "borderline" as the "rating" intruction suggested.

**Limitations:**

yes

**Quality:**

3

**Strengths And Weaknesses:**

Strengths:
- The use of reasoning model for routing is interesting yet intuitive, utilizing the more powerful reasoning paradigm on the model routing use case. The strong generalization may suggest the model learns some useful insights for routing the models.
- The reward design enables successful RL training without suffering from reward hacking. Specifically, the authors choose to use rewards with verifiable rules and uses a hierarchical reward system.
 - Extensive evaluation is done on a list of benchmarks and baseline methods.

Weakness:
- The multi-round, iterative nature of Router-R1 introduces increased inference latency due to the reasoning steps. Depending on the task, this may be negligible or significant. An analysis on this would help improve the paper.
 - The application shown here are generally QA tasks. A complex routing method may work well on complex scenarios (such as SWE where the router works like a PM). It would be interesting to see application in similar domains.

---

> ### Author Rebuttal · Authors · 2025-07-30
>
> Thanks for your valuable feedback and appreciation of the quality, clarity, and contribution to our work. We appreciate your insights and would like to further address the concerns you raised about our work.
>
> ## **Inference Latency**
>
> We thank the reviewer for the insightful and constructive comment.
> We would like to clarify a key point: **the multi-round, iterative reasoning of Router-R1 does not necessarily lead to increased inference latency.** Instead, it adapts the number of reasoning steps to the difficulty of each task, which can actually reduce latency for simpler questions and avoid unnecessary computation.
>
> **(1) Adaptive reasoning reduces latency for simple tasks**
>
> As correctly noted by the reviewer, the latency impact depends on the task complexity. We conduct a detailed analysis in **Figure 2a, Section 5.4 (Page 9)**, where we measure the average number of LLM calls across seven QA benchmarks. In comparison with single-hop benchmarks, multi-hop tasks (e.g., Musique, 2WikiMultiHopQA) require more reasoning steps. This task-dependent adaptivity helps minimize latency where possible. (**Lines 315–322**)
>
> **(2) Router-R1 offloads reasoning to lightweight models**
>
> Router-R1 can reduce latency by delegating parts of the reasoning process to smaller, faster models in the routing pool, instead of relying solely on the policy model. This is particularly beneficial when the policy LLM is larger than some candidate models (a setting that is increasingly relevant in practical deployments).
> Although in our current setup the policy LLM is relatively lightweight, the design of Router-R1 naturally supports scalable offloading, where a larger central controller coordinates a pool of efficient LLMs, potentially reducing the end-to-end latency compared to monolithic processing by a single large model.
>
> **(3) Case studies demonstrate latency-efficient routing behavior**
>
> In Appendix B (Page 22), we present case studies showing that with cost-aware RL training, Router-R1 prefers small models when possible, escalating to larger ones only when necessary. While the main optimization objective is cost, this behavior often correlates with lower latency, since smaller models respond faster in practice.
>
> **(4) Configurable cost-performance tradeoff**
>
> We further provide a systematic study of different cost coefficients $\alpha$, showing that Router-R1 allows flexible tuning between performance and efficiency. While this primarily reflects monetary cost, lower cost settings often result in fewer or faster model calls, which can also benefit latency.
> (The “cost” in the table denotes unnormalized raw cost rewards)
> | **Methods**                | **NQ EM↑** | **NQ Cost↓** | **PopQA EM↑** | **PopQA Cost↓** | **HpQA EM↑** | **HpQA Cost↓** | **2wiki EM↑** | **2wiki Cost↓** |
> | -------------------------- | ----------- | ------------- | ------------- | --------------- | ------------- | --------------- | ------------- | --------------- |
> | **Qwen2.5-3B-Instruct**    |             |               |               |                 |               |                 |               |                 |
> | Prompt LLM                 | 0.300       | 20.0          | 0.340         | 17.6            | 0.268         | 20.1            | 0.262         | 20.2            |
> | Largest LLM                | 0.296       | 20.2          | 0.354         | 17.6            | 0.278         | 20.1            | 0.274         | 20.1            |
> | KNN Router                 | 0.262       | 41.0          | 0.222         | 30.3            | 0.224         | 45.4            | 0.196         | 51.8            |
> | MLP Router                 | 0.252       | 35.8          | 0.222         | 21.0            | 0.198         | 29.5            | 0.210         | 41.3            |
> | BERT Router                | 0.230       | 26.0          | 0.192         | 16.3            | 0.216         | 26.0            | 0.206         | 26.3            |
> | RouterDC                   | 0.278       | 57.5          | 0.282         | 21.8            | 0.244         | 62.3            | 0.218         | 40.7            |
> | GraphRouter                | 0.276       | 29.6          | 0.280         | 19.2            | 0.234         | 24.7            | 0.180         | 28.6            |
> | Prompt LLM\*               | 0.258       | 286.4         | 0.256         | 111.7           | 0.206         | 313.4           | 0.248         | 222.4           |
> | KNN Router\*               | 0.236       | 102.2         | 0.232         | 49.8            | 0.154         | 133.0           | 0.234         | 99.4            |
> | **Router-R1-Qwen (α=0.0)** | **0.384**   | 149.4         | 0.391         | 97.5            | **0.313**     | 130.4           | **0.393**     | 145.5           |
> | **Router-R1-Qwen (α=0.6)** | **0.384**   | 150.3         | **0.421**     | 75.1            | 0.305         | 123.7           | 0.382         | 113.4           |
> | **Router-R1-Qwen (α=0.7)** | 0.314       | 31.4          | 0.285         | 16.9            | 0.236         | 25.9            | 0.214         | 31.2            |
> | **Router-R1-Qwen (α=0.8)** | 0.317       | 28.7          | 0.275         | 14.7            | 0.221         | 28.5            | 0.183         | 30.1            |
> | **Router-R1-Qwen (α=0.9)** | 0.241       | **5.4**       | 0.238         | **5.3**         | 0.176         | **5.4**         | 0.228         | **6.3**         |
>
> **(5) System-level latency can be minimized in practice**
>
> While API-based access to external models (as used in our experiments) may introduce network overhead, this is not inherent to Router-R1. In practical deployment, models can be hosted locally or optimized via standard acceleration techniques. System-level latency engineering is orthogonal to the algorithmic contributions of this work and is left to future work.
>
> **In summary, Router-R1 does not introduce unnecessary latency, but rather provides a task-adaptive mechanism that adjusts inference effort based on question complexity. This design enables a flexible and efficient reasoning strategy that balances latency and accuracy.**
>
> ## **Application in Similar Domains**
>
> We thank the reviewer for the thoughtful and encouraging suggestion. While our current experiments focus on QA tasks, the modular and hierarchical design of Router-R1 is general and well-suited to complex domains such as SWE, where the router can coordinate submodules like a project manager. We are excited to explore such applications in future work, and we believe Router-R1 holds strong potential for such applications.
>
>
> ## **The cost of policy LLM**
>
> We appreciate the reviewer’s thoughtful suggestion. In our current setup, we use a relatively small 3B policy LLM, whereas the candidate models are significantly larger.
> Moreover, as shown in Appendix B (Case Study), the vast majority of output tokens come from the candidate LLMs. The policy LLM primarily acts as a controller, assigning subtasks and summarizing information, with a minor computational footprint. For this reason, we did not include it in the cost calculation.
> That said, we agree this is a valid and practical extension, especially in industrial settings where larger policy models may be used. In such scenarios, it is indeed feasible to integrate the policy LLM cost into the overall cost analysis or even let Router-R1 jointly optimize for the total cost.
> We view this as a valuable direction for scaling Router-R1 beyond the academic setting and plan to explore it in future work.
>
>
>
> Thank you again for your constructive feedback, and we look forward to further engaging in discussions to improve the quality and impact of our work.

---

> > ### Comment · Reviewer_iXcu · 2025-08-03
> >
> > Thanks for the detailed response. I would keep my current scores.

---

> > > ### Author Response · Authors · 2025-08-04
> > > **Response to Reviewer iXcu**
> > >
> > > Thank you very much for your time and thoughtful evaluation. We truly appreciate your constructive feedback and support.

---

### Author Response · Authors · 2025-08-09
**General Response**

We appreciate the constructive feedback provided by the reviewers and their recognition of our work. The reviewers acknowledged that our work is novel, meaningful, sound, and holds practical application value overall. In particular, the reviewers appreciated the design of Router-R1's training objective (especially the cost rewards), which equips the model with the ability to adaptively and dynamically balance the cost–performance trade-off, thereby enhancing its practical value in real-world applications. Additionally, our comprehensive experimental design and the state-of-the-art results we achieved are also appreciated by the reviewers, which are also highlights of our work.

In response to the concerns raised, we have provided very detailed and comprehensive explanations and clarifications. For the experiments, we added a performance–cost trade-off analysis table for Router-R1, along with a cost analysis comparing the baseline and Router-R1, further highlighting the superiority of our approach. We also conducted an additional sensitivity analysis of Router-R1 to different prompts, supplemented with corresponding baselines, which further demonstrates the robustness of our proposed method. In addition, we offer sufficient clarifications and share new insights to further address the reviewers’ concerns and reaffirm the novelty and contribution of our work. Finally, we also clarified the design of Router-R1’s training objective to resolve the reviewers’ minor concerns.

We respectfully request that the reviewers consider these improvements based on the constructive feedback provided. We believe these enhancements strengthen the paper and provide valuable insights to the community, and we hope for consideration of increasing the scores of our submission. Thank you for your thoughtful consideration.

---

### Note · Authors · 2025-08-15

We sincerely thank all the reviewers for their constructive feedback and recognition of our contributions.

The reviewers acknowledged that our work is novel, meaningful, sound, and of strong practical value (*iXcu, Gukk, f8xo*). In particular, they appreciated the design and multi-round routing of Router-R1, especially the cost rewards, which enable adaptive, dynamic balancing of the cost–performance trade-off, enhancing its value in real-world deployments (*iXcu, Gukk, 7Web, f8xo*). They also recognized the comprehensiveness of our experiments and the state-of-the-art results achieved (*iXcu, Gukk, 7Web*).

In response to the main concerns, we have provided clarifications and additional experiments:

- **Clarification and new insights about Router-R1’s motivation, novelty, and objective design** – We further clarified and emphasized the motivation and novelty of Router-R1 and explained the rationale behind the cost rewards and how they enable adaptive cost–performance balancing (*7Web, f8xo*).

- **Missing baselines** – Added additional baselines for comprehensive comparisons (*Gukk*).

- **New candidate models** – Analyzed the impact of adding new models to the routing pool, demonstrating flexibility (*7Web*).

- **Performance–cost trade-off** – Added a trade-off analysis table and compared cost rewards against baselines, further supporting the robustness and applicability of our approach (*iXcu, Gukk, 7Web, f8xo*).

We will also refine the revised version to improve clarity and broaden applicability: updating experiments with expanded analyses and baselines, revising statements for precision, and adding discussions on scalability, prompt sensitivity, and integration of new models.

Finally, we greatly appreciate that several reviewers view our work as promising and potentially inspiring further research. We believe our framework not only offers a novel, practically valuable routing method but also opens new directions for adaptive cost–performance trade-offs in large-scale model deployments.

---

### Decision · Program_Chairs · 2025-09-17

**Decision:**

Accept (poster)

**Comment:**

This paper proposes Router-R1, a RL framework for LLM routing and aggregation. Unlike single-round routers that assign a query to a single model, Router-R1 formulates this as a sequential decision-making process. The router itself is a reasoning LLM that alternates between internal deliberation and external model calls. It learns a policy to select the most suitable model for a given sub-task, integrating the response into its context to progressively build a final answer. The training is guided by a novel, hierarchical reward function that balances performance (answer quality), format, and cost. Experiments on seven QA benchmarks demonstrate that Router-R1 outperforms strong baselines and provides a flexible, cost-aware trade-off.

However, a number of significant weaknesses were raised, including:
- Reviewers noted missing baselines, a lack of runtime efficiency analysis, and a need for clearer Pareto curves to illustrate the cost-performance trade-off.
- One reviewer argued the method is not fundamentally different from existing tool-learning frameworks.
- The paper's experiments are limited to a small 3B router model, and it is unclear how the approach would perform with larger policies or with different parameter-efficient tuning methods.
- The multi-round, iterative nature of the method may introduce significant latency and cost, which the authors' analysis does not fully address.

The authors provided rebuttals to address these concerns.
- The authors added results for an expanded analysis of the cost-performance trade-off across different alpha values. They also added results for individual models to contextualize the trade-offs
- The authors clarified that Router-R1 is distinct from tool-learning methods like Search-R1 because it learns to route among semantically similar LLMs. They emphasized that the core novelty lies in applying RL to unify multi-round reasoning with cost-aware routing.
- The authors argued that Router-R1's adaptive nature can reduce latency for simpler tasks, though they acknowledge that direct wall-clock runtime comparisons were not possible due to their experimental setup. They also noted that the cost of the small 3B router model is minimal compared to the larger models in the pool.

The paper presents a original idea with a clear design and reasonable empirical results. The authors' rebuttals were thorough and responsive, addressing most of the reviewers' concerns with additional experiments and insightful clarifications. The introduction of cost-aware RL for LLM routing is a practical contribution. While some concerns about efficiency and a lack of certain comparisons remain, they are not severe enough to warrant rejection. The potential for this work to inspire future research in multi-round, cost-aware LLM routing makes it a valuable contribution to the NeurIPS community. The work's strengths outweigh its weaknesses.